

# Simulating the recent drought-induced mortality of European beech (*Fagus sylvatica L.*) and Norway spruce (*Picea abies L.*) in German forests

Gina Marano[1], Ulrike Hiltner[1], Nikolai Knapp[2], Harald Bugmann[1]

[1] Forest Ecology, Institute of Terrestrial Ecosystems, Department of Environmental Systems Science, ETH Zurich, 8092 Zürich, Switzerland
[2] Thünen Institute of Forest Ecosystems, 16225 Eberswalde, Germany

*Correspondence to*: Gina Marano (gina.marano@usys.ethz.ch)

**Abstract.** Drought is increasingly recognized as a critical driver of forest dynamics, altering tree species' growth, dominance and survival. To better understand these dynamics, we used a process-based modeling approach to investigate drought-related mortality of European beech (*Fagus sylvatica L.*) and Norway spruce (*Picea abies L.*) in German forests. The predisposing-inciting (PI) framework for drought-induced tree mortality incorporated in ForClim v4.1 was combined with a bark beetle module for Norway spruce to account for a key contributing factor, leading to ForClim v4.2.

Our study addressed four hypotheses: (1) the PI framework, initially developed for Swiss beech forests, is effective across the broad ecological and climatic gradients found in Germany; (2) Soil properties, namely soil water holding capacity (AWC) and soil heterogeneity, have a strong influence on drought-related mortality, complementing climatic drivers; (3) local soil heterogeneity modulates drought-related mortality by amplifying mortality risk through limited microsite variability, or dampening it by providing moist refugia; (4) incorporating bark beetle damage ameliorates model performance for simulating drought-related mortality of Norway spruce. Our modelling approach deliberately forgoes calibration to better investigate the underlying mechanisms and drivers of drought-induced tree mortality.

We conducted simulations across hundreds of plots of the ICP Forest Level I network in Germany, covering a wide gradient of climate and soil conditions. ForClim reproduced the general patterns of drought-related mortality, highlighting the ability of the PI framework to capture emergent mortality patterns across a range of environmental conditions. However, mismatches in magnitude and trends highlight areas for improvement. Discrepancies were attributed to sparse mortality data, the drought sensitivity of the bark beetle submodule, and the absence of regional calibration. Our results revealed the critical role of AWC and local soil heterogeneity in modulating drought responses. Sites with low AWC experienced significantly higher mortality rates, while high AWC provided a buffering effect, bringing simulated outcomes closer to observed data. Furthermore, soil heterogeneity played a mitigating role, with sites exhibiting uniform soils showing higher mortality risk, thus emphasizing the importance of the spatial variability of soil properties for dampening drought impacts. Lastly, the new bark beetle submodel, even though highly simplified, considerably improved the simulation of drought-related mortality patterns in Norway spruce-dominated sites.



This study underscores the value of process-based models like ForClim for disentangling the mechanisms underlying forest vulnerability and drought-induced mortality. However, improvements such as finer-resolution mortality and crown condition data, as well as regional model calibration, would be useful to enhance its predictive accuracy. Our findings contribute to the

better understanding, forecasting and managing forest resistance under current and future climatic conditions.

*Keywords*: drought-induced mortality, bark beetle, forest gap model, ForClim, soil water holding capacity.

## 1 Introduction

In recent decades, the frequency and intensity of drought extremes increased markedly, becoming a defining feature of the

global climate regime (Caretta et al., 2022). Intensified drought events pose considerable threats to forest ecosystems world-wide, disrupting forest dynamics and altering species dominance (Cavin et al., 2013; Fensham et al., 2015). European beech (*Fagus sylvatica L.*) and Norway spruce (*Picea abies L.*) historically were cornerstone species in European forestry due to their economic and ecological value (FOREST EUROPE and FAO, 2020). However, the recent prolonged and intense droughts exposed unprecedent vulnerabilities in these species (Bonannella et al., 2024; Kasper et al., 2022; Schuldt et al., 2020). Scien-

tists, forest managers and nature conservationists in these areas are debating the future viability of both species, emphasizing the need for adaptive management strategies that consider species-specific responses to drought (Bottero et al., 2021; Meyer et al., 2022; Vitali et al., 2017). This is particularly evident in lowland regions such as Germany, where drought has been a major cause of decline in recent years (Langer and Bußkamp, 2023; Spiecker and Kahle, 2023; Thonfeld et al., 2022).

In this context, it is of paramount importance to understand the conditions that led to the observed decline and mortality

patterns, so as to ultimately be able to predict the behavior of these tree species under further anthropogenic climate change. To this end, systematic surveys like the International Co-operative Programme on Assessment and Monitoring of Air Pollution Effects on Forests (ICP Forests) provide valuable data over large areas (Gazol and Camarero, 2022; George et al., 2022). They include crown defoliation assessments, which have shown promise in detecting early signs of stress that is predisposing trees to die (Hunziker et al., 2022). Monitoring data such as those from ICP Forests were used to statistically model tree mortality

probability (Anders et al., 2025; Camarero, 2021; Knapp et al., 2024). While valuable for assessing and mapping forest health and detecting early signs of stress, such statistical models offer limited insight into the underlying processes driving observed mortality patterns, are difficult to extrapolate to future conditions, and, being closely tied to underlying empirical data, often lack transferability across regions and tree species.

Process-based dynamic vegetation models (DVMs) provide a complementary approach by simulating tree mortality on a more

mechanistic basis, incorporating complex interactions between climate, soil, and tree physiology (Bugmann et al., 2019). However, DVMs continue to struggle with accurately reproducing observed patterns of forest response to drought, particularly due to an incomplete understanding of the ecophysiological processes underlying tree mortality (Fischer et al., 2025; Kolus et al., 2019; Trugman, 2022), but also uncertainties associated with soil properties (Camarero, 2021; Kolus et al., 2019; Trugman



et al., 2021). Uncertainties in our understanding of soil properties and their role in mediating drought responses make it chal-
lenging to accurately assess how soil moisture deficits shape the long-term drought response, either buffering or amplifying
atmospheric drought signals (Koster et al., 2009; Trugman et al., 2018; Ukkola et al., 2016).

Recent research has suggested a path forward by integrating predisposing and inciting mortality factors in DVMs (hereafter
PI scheme). The PI scheme integrated in the ForClim model v4.1 (Marano et al., 2025) features a 'drought memory' term as a
predisposing factor, reflecting the species-specific effects of prolonged drought. Additionally, it includes inciting factors such
as drought duration and seasonal soil moisture deficits, which may hinder bud break in spring, cell division and expansion in
summer, and carbon reserve buildup in fall for the following year. The approach highlights how consecutive dry years and soil
moisture deficits can amplify mortality risk, even though processes such as hydraulic failure and carbon starvation are not
modelled explicitly (Liu et al., 2021; Yao et al., 2022). Marano et al. (2025) found that ForClim v4.1 was able to simulate the
recent drought-induced tree mortality at six mesic beech sites in Switzerland as well as at a xeric Scots pine site in the conti-
nental Alps while providing a realistic depiction of potential natural vegetation from the cold to the dry treeline in Europe.
That same model was also found to be able to realistically simulate the recent massive wave of Norway spruce mortality at
one site in the Harz mountains in Germany (Fischer et al. 2025).

Yet, the generality of the new formulation for drought-induced mortality remains unknown because the ability of ForClim
v4.1 to simulate the recent observed mortality patterns over heterogeneous regions rather than at individual sites remains
untested. Furthermore, the model does not currently account for key contributing factors such as bark beetle infestations, which
are known to play a pivotal role in driving mortality under drought (Hlásny et al., 2021a; Netherer et al., 2019). Several studies
highlighted that biotic stressors often interact with abiotic factors, amplifying tree vulnerability and mortality rates, especially
in regions prone to prolonged droughts, for instance in the Swiss Alps (Bigler et al., 2006; Dobbertin et al., 2004; Rigling and
Cherubini, 1999). Incorporating these contributing factors in the PI scheme, hence, to be named PIC scheme, would not only
enhance its predictive accuracy but also improve our understanding of the complex dynamics governing drought-related tree
mortality, and our predictive ability under changing climatic conditions.

In this study, we aimed to evaluate the generality of the PI scheme of Marano et al. (2025) beyond the site-specific tests
reviewed above and developed it further into a PIC framework. We focused on European beech and Norway spruce due to the
prevailing uncertainty regarding their response to future climatic extremes. Taking into account that soil properties may play
a major role for modulating drought-induced mortality, we decided to address the following research questions:

1. Can the PI scheme predict drought-induced mortality across diverse environmental conditions?
2. How strongly is soil water holding capacity and its small-scale spatial heterogeneity modulating drought-induced tree
   mortality of European beech and Norway spruce, in addition to the effects of climate variability?
3. Does the integration of a simple model for bark beetle activity (PIC scheme) enhance the predictive ability of the forest
   model in the context of Norway spruce mortality?



## 2 Material and Methods

### 2.1 Site selection and mortality observations

The ICP Forest Level I plots, in Germany also known as crown condition survey ("Waldzustandserhebung", hereafter WZE), constitute a large network in which tree vitality and mortality are observed annually for 24 dominant trees per plot, attributing

causes to the mortality events as far as possible (Figure 1, panel B). Overall, the WZE contains about 410 plots on a regular 16 km x 16 km grid across Germany (Wellbrock et al., 2018). On each plot, crown defoliation is being evaluated for the same sample trees year after year. Since 1998, the likely reasons are recorded for trees that have died. Only natural causes that kill trees and leave them standing are considered, i.e., harvested trees and windthrows are excluded. A tree is labelled as dying when it has been alive in the previous year and is either standing dead with fine twigs still present but 100% defoliation (tree

status 0 or 9 according to Wellbrock et al., 2018) or leaving the WZE sample collective as a dead tree with a natural cause of death, including salvage logging (tree status 12, 32, 33, 34, 42 or plot status 12).

We selected all sites of the ICP Forest Level I plot network in Germany (Figure 1, panel A) where progressive crown decline (i.e. crown defoliation) associated with drought-related stress of the respective species (European beech and Norway spruce) was observed. This amounted in total to 30 plots dominated by beech and 119 plots dominated by spruce. We used only those

trees that were classified as having died due to drought-related factors and processes (both biotic and abiotic). The overall number of drought-related dead trees in the WZE from 1998 to 2022 was merely 51 for *Fagus sylvatica*, but for *Picea abies* it was 1,110 (cf. Supplementary Material, Figure S1.1-S1.2), whereas 47,621 and 82,551 living trees were monitored repeatedly, respectively (Knapp et al., 2024). This imbalance in the observed mortality data, particularly for beech, may limit the representativeness of the dataset and its ability to capture the full range of drought-induced mortality events. To address this limi-

tation, our study employed a combination of model simulations and plausibility analyses to evaluate the potential influence of soil moisture and its spatial heterogeneity on drought-induced mortality.





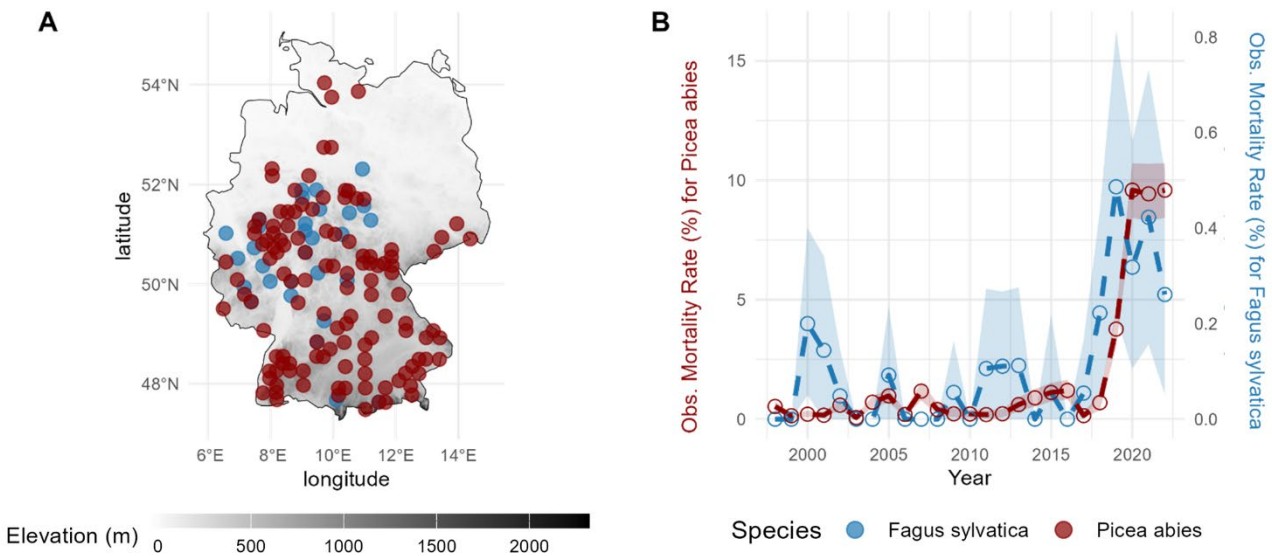

**Figure 1**. Location of the ICP Forest Level I plots for the present study (**A**) and the respective observed annual mortality rates (**B**) for European beech (blue) and Norway spruce (dark red) averaged across sites. The shaded blue and red areas represent
±1 standard deviation derived via a bootstrap procedure to show the variability in the observed data.

## 2.3 ForClim Model

### 2.3.1 Brief overview

ForClim is a forest gap model that was originally designed to capture the dynamics of temperate mountain forests in Central Europe and account for climate change (Bugmann, 1994; Bugmann 1996). It allows for both short- and long-term estimations of forest dynamics that are simulated on small areas ('patches'), normally with a size of 1/12 ha (ca. 800 m$^2$), each representing one out of many stochastic realizations of tree population dynamics that are spatially independent from each other. The simulation results from many patches (typically 200) that are averaged to obtain forest dynamics at the stand scale.

ForClim is based on a minimum number of ecological assumptions to capture the influence of climate on ecological processes, i.e. establishment, growth, competition, and mortality of trees, in the context of forest dynamics. Environmental constraints, particularly temperature and soil water availability, influence tree-level performance, thereby regulating growth and increasing the risk of mortality. Each species is defined by a suite of functional traits including shade tolerance, frost resistance, and drought resistance, among others, which determine how it responds to environmental conditions. Model outputs are calculated annually for each tree, enabling both fine-grained analyses at the individual level and assessments of species composition and stand dynamics.





We employed ForClim v4.1 (Marano et al., 2025) which builds upon ForClim v4.0.1 (Huber et al., 2020). While it keeps a modular approach, it presents distinct features. First, v4.1 uses time series of monthly weather data (precipitation sum, mean temperature) to derive bioclimatic variables in the *weather* submodel, rather than employing a random weather generator operating at the patch scale, as done in previous model versions. ForClim's *water* submodel builds on the Thornthwaite and Mather soil water balance (Bugmann and Cramer, 1998), following a mono-layer bucket model approach, i.e. a single soil

layer is conceptualized as receiving and storing all incident water until its capacity is reached. Bucket models use the concept of "bucket size" (kBS), representing the maximum plant-available water in the soil, thus corresponding to the available water capacity (AWC).

    To account for the fine-grained spatial heterogeneity of soil properties in forests, in ForClim v4.1 distinct soil water properties (AWC) are assigned to each forest patch. A lognormal probability distribution is used to represent the patch-level variability

in AWC, reflecting the skewed and stochastic nature of these properties. This distribution is parameterized using the mean and minimum values of the parameter kBS (AWC), which determine the standard deviation ($\sigma$, Eq. 1) and mean ($\mu$, Eq. 2) of the lognormal distribution (for details cf. Marano et al., 2025).

$$\sigma = \ln\left(\frac{kBS_{mean}}{kBS_{min}}\right) \qquad (1)$$

$$\mu = \ln(kBS_{mean} - kBS_{min}) - \frac{\sigma^2}{2} \qquad (2)$$

This heterogeneity among patches is particularly relevant for assessing the impacts of drought extremes on the soil water balance in forest ecosystems. Specifically, heterogeneous soils may provide microsites with high levels of water retention, acting as potential drought refugia for plants (Gazol et al., 2018; Kirchen et al., 2017; Ripullone et al., 2020). Moreover, because $\sigma$ quantifies the spread in the lognormal distribution of AWC, it serves as a direct indicator of the degree of heterogeneity in water availability within a forest stand.

Following Manion's (1981) Decline-Disease Theory, in the *plant* submodel of ForClim v4.1 predisposing and inciting factors are distinguished that contribute to tree mortality, as derived and explained in detail by Marano et al. (2025). Overall, the probability of stress-induced mortality in ForClim v4.1 emerges from three key elements: *(i)* a general carbon memory component, reflecting tree vulnerability arising from persistent sub-optimal growth; *(ii)* drought-related predisposing factors, represented by a drought memory that is related to the species-specific drought tolerance; and *(iii)* inciting factors, arising from

cumulative drought stress (via the seasonal ratio of supply and demand for soil water, which is closely related to Vapor Pressure Deficit) in combination with soil moisture shortage in spring and fall.

    This framework acknowledges that a series of dry years can increase mortality risk under prolonged summer and early- or late-season soil moisture deficits, conditions that are closely related to carbon starvation (long-term effect) and the onset of hydraulic failure (short-term effect), although neither of these physiological processes is modeled explicitly. In ForClim v4.1,

the predisposing and inciting factors *sensu* Manion give rise to a constant stress-induced mortality probability (cf. Marano et



al. 2025). However, contributing factors such as biotic agents (e.g., insect outbreaks, fungal pathogens) are not accounted for in ForClim v4.1.

### 2.3.2 A simple spruce bark beetle submodel

An empirically based formulation for estimating bark beetle (mainly *Ips typographus* L.) disturbance probability for Norway spruce was derived to add contributing factors (C) to the PI scheme of ForClim v4.1 (Marano et al., 2025; Manion 1981). This was necessary since drought-related spruce mortality is often amplified by bark beetle outbreaks. The simple bark beetle sub-model consists of a base probability for a beetle outbreak, which is composed of (1) a base annual probability, (2) a flag for a potential year of infestation, and (3) an inciting drought stress term, as explained below.

### 2.3.2.1. Base annual probability

The annual base probability for a bark beetle outbreak (*Pbark*) was derived from the theoretical probability of bark beetle disturbance in spruce-dominated forests (Hlásny et al. 2021, their Fig. 4). We adapted the disturbance map from Hlasny et al. (2021) to obtain a base probability of bark beetle outbreaks for Germany under the climate of 1979-1990 (cf. Hlasny et al., 2021, their Figure 4 and Appendix 2). We converted the original six qualitative classes for the outbreak probability (i.e., *No Spruce, Very Low, Low, Medium, High,* and *Very High*) into quantitative values from 0 to 100%. Our classification was de-
signed to ensure that the lower outbreak probability classes (i.e., *Very Low* and *Low*) had smaller ranges, while the higher outbreak probability classes (i.e., *Medium, High,* and *Very High*) reflected increasing values with smooth transitions (Tab. 1). This heuristic approach we used combined fixed percentages for the lower classes and an exponential progression for the higher classes, as explained in Supplementary Material SM 2.2. To obtain the base probability of bark beetle outbreaks (*Pbark*) for the area of Germany, we averaged all pixel values in Hlasny et al. (2021), resulting in a value of *Pbark* = 20%.

**Table 1.** Base annual probability for the six bark beetle outbreak classes with estimated ranges and midpoints. The original class definitions are derived from Hlasny et al. (2021).

| Class | Range (%) | Midpoint (%) |
|---|---|---|
| *No spruce* | 0 | 0 |
| *Very Low* | 0-5 | 2.5 |
| *Low* | 6-15 | 10.5 |
| *Medium* | 16-34 | 25 |
| *High* | 35-65 | 50 |
| *Very High* | 66-100 | 83 |




### 2.3.2.2. Potential year of infestation

To identify a potential year of infestation, we assessed the estimated number of bark beetle generations within that year ($g_{Gen}$) against the stress status of trees based on the intensity of drought within that year. The number of bark beetle generations (Eq. 3) per year are calculated as a function of the annual degree-day sum ($mDDAn$) based on the relationships from Jakoby et al. (2019).

$$g_{Gen} = \begin{cases} 0, & mDDAn \leq 400 \\ \left(\frac{mDDAn-400}{1600}\right), & 400 < mDDAn \leq 2000 \\ 2, & mDDAn \geq 2000 \end{cases} \tag{3}$$

### 2.3.2.3. Inciting factor for drought stress

To account for the fact that drought stress weakens spruce trees, we considered a drought-related sensitivity threshold ("drought tolerance", $kBeetle_{DrTol}$) based on the species-specific drought tolerance parameter ($kDrTol$, Bugmann and Cramer, 1998) scaled by a factor of 2/3, in order to allow moderate drought events to have a noticeable effect on spruce vigor (Eq. 4).

$$kBeetle_{DrTol} = kDrTol \cdot \frac{2}{3} \tag{4}$$

The overall probability of tree mortality induced by bark beetles ($PDist$; Eq. 5) was then determined by the interplay of biotic factors (i.e., the number of bark beetle generations $g_{Gen}$) and environmental stressors (i.e., annual drought stress; $mDrAn$) that identify a potential year of bark beetle infestation, and finally the predisposing factor of the drought memory ($uDrM$) (for details, see Marano et al. 2025):

$$PDist = \begin{cases} Pbark, & (g_{Gen} > 1.5) \wedge (mDrAn > kBeetle_{DrTol}) \wedge (uDrM > 1) \\ 0, & else \end{cases} \tag{5}$$

This new bark beetle model led to ForClim v.4.2, which incorporates predisposing, inciting as well as contributing mortality factors *sensu* Manion (1981) (PIC scheme).

### 2.3.4. Simulation settings

To initialize the ForClim model, stand-level information for the ICP Forest Level I plots was required, but each plot provides just observations for 24 dominant trees with no specified plot size or scaling to the hectare. Therefore, we derived an approximate stand-level inventory for the year 2000 defined as the equilibrium between current climate, soil properties, and forest vegetation. To do so, we ran single-species simulations from bare ground for 2000 years. We applied this procedure for each of the selected ICP Forest Level I plots, using 200 patches with a size of 800 m$^2$ to account for stochasticity. Across all plots, nitrogen availability ($kAvN$) was set to non-limiting 180 kg ha$^{-1}$ year$^{-1}$, and we assumed flat terrain ($kSlAsp = 0$). For each plot, we retrieved interpolated climate data from the German Weather Service (DWD) at 1 km resolution (Kaspar et al., 2019).



Precipitation ranges from 549 to 2063 mm for the European beech plots and from 655 to 2182 mm for the Norway spruce plots (cf. Figure SM S1.1-S1.2). Annual mean temperature of the plots ranges from 5.3 to 10.5 °C for both species. All climate data

refer to the period 1950-2023.

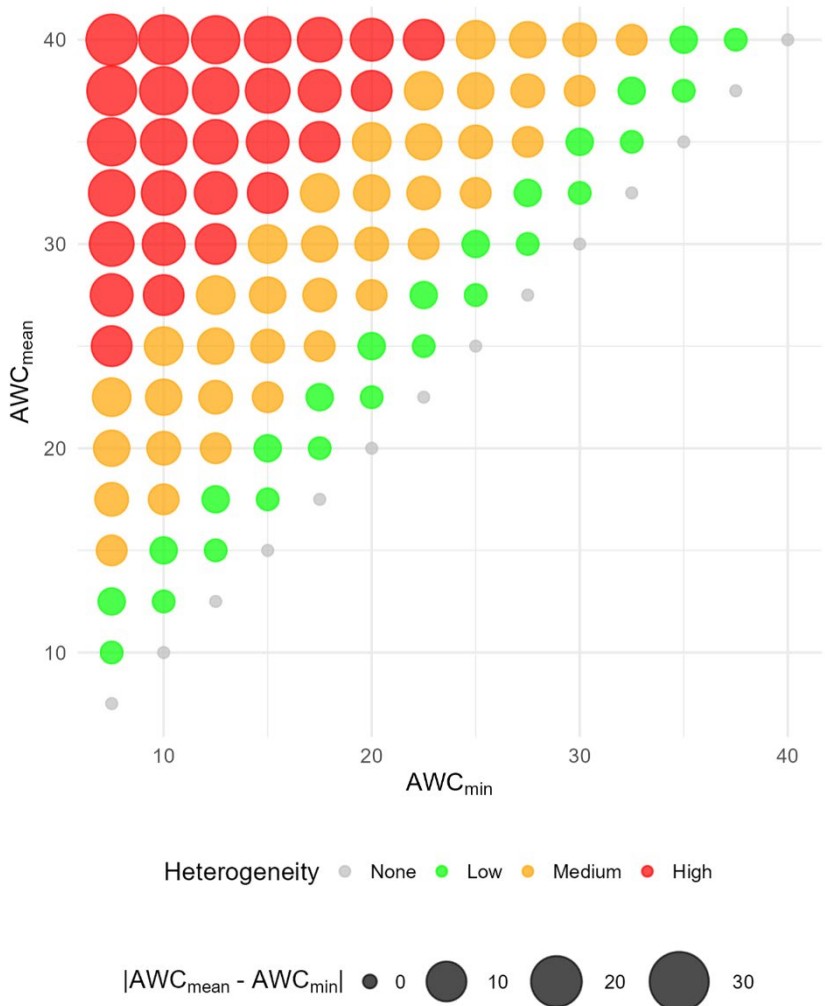

**Figure 2.** Soil scenarios defined as combinations of minimum and mean AWC. Distinct colors indicate the soil heterogeneity classes, ranging from no heterogeneity ($AWC_{min} = AWC_{mean}$, *None*), across low and medium to high heterogeneity classes.

We attempted to estimate the minimum and mean values of soil AWC down to rooting depth from maps of the Bundesanstalt

für Geowissenschaften und Rohstoffe (BGR) at 500 m spatial resolution (Meinel, (2015), cf. Figure SM 2.1.1). Yet, the re-

sulting soil properties were deemed questionable, as they would for example posit that European beech thrives on soils with





an AWC <50 mm (cf. Figure SM 2.1.2), which is ecologically implausible (Kirchen et al., 2017; Pichler et al., 2009; Walthert et al., 2021a). Thus, we decided to evaluate the likely site-specific AWC and its spatial heterogeneity through sensitivity analysis by defining a range of scenarios (Marano et al., 2025). We set the ranges of the minimum and mean values of AWC

based on the BGB maps (Figure SM 2.1) but considered additional, ecologically relevant values to cover a wider range of potential soil conditions. The resulting scenarios spanned $AWC_{mean}$ values from 7.5 to 40 cm, covering the range found in the BGB maps and including more extreme values to assess the sensitivity of the model to soil water availability. The absolute difference between $AWC_{min}$ and $AWC_{mean}$ represents a gradient of plot-level soil heterogeneity classes which we assigned to four distinct groups (Figure 2). To quantitatively constrain the soil heterogeneity classes, we also calculated the corresponding

sigma (σ) of the lognormal distribution for each soil moisture scenario (cf. Table S 2.1). A total of 105 AWC scenarios were tested to evaluate two hypotheses: *(a)* under conditions of low precipitation, a low mean available water capacity ($AWC_{mean}$) induces high mortality rates for European beech and Norway spruce, indicating that in Germany these species predominantly thrive on more favorable soils, and *(b)* local spatial heterogeneity in soil water availability, represented by the plot-level difference between $AWC_{mean}$ and $AWC_{min}$, influences tree survival by (i) exacerbating drought-induced mortality when $AWC_{mean}$

is low and spatial heterogeneity is also low, or (ii) providing moist refugia in highly heterogeneous soils. This approach allows for a more realistic exploration of ecologically sound AWC values, yet their patterns are still based upon mapped products.

For each ICP Forest Level I plot, the ForClim model was initialized with the simulated equilibrium (the state of the forest structure – i.e., species, DBH and height of all trees – in the simulation year 2000), and simulations were run from 2000 until 2022. A total of 31'890 simulation scenarios (states · sites · soil scenarios · species) were run on the Euler High Performance

Computing Cluster of ETH Zürich. These simulation results were filtered for trees with DBH ≥ 40 cm so as to better match the simulation results with the observations, which only include 'dominant' trees. Yet, the simulation results contain all causes of growth-related mortality and thus are likely to overestimate the observed data. Therefore, we subtracted the simulated "background mortality" rates (which are independent of growth patterns in the model) from the simulated values. They correspond to an annual mortality rate of 0.921% and 0.495% for European beech and Norway spruce, respectively (Forest Ecology

ETH Zürich, 2019). Notably, in contrast to the observational data, which focus on drought-related mortality, the simulated mortality rates still contain mortality events that are due to other stressors (e.g., light availability), and thus simulated mortality rates are likely to remain higher than the observed ones.

In this simulation study, no model calibration was performed, as we aimed to assess the model's ability to identify the drivers (PIC and PIC factors) shaping the system response, rather than to fine-tune parameters to best match observational data. Our

approach ensures that the results reflect intrinsic model behaviour rather than adjustments made to improve fit, aligning with previous studies that recommend to integrate process-based mechanisms in DVMs rather than empirical calibration (McDowell et al., 2011; Meir et al., 2015).



**2.4 Statistical analyses**

The observed mortality rate ($M_{obs}$) from the ICP Forest Level I data was calculated by dividing the number of dead trees by
the number of living trees from the previous year (normally, $n = 24$ for each year at each plot), as done by Knapp et al. (2024).
The results were averaged across all plots for each species. Similarly, the simulated mortality rate ($M_{sim}$) of the large trees
(DBH > 40 cm) was averaged across all simulated sites (each consisting of 200 patches) for each species.

The performance of the ForClim model for each AWC scenario was evaluated by employing the mean absolute error ($MAE$),
the root mean squared error ($RMSE$) and the coefficient of determination (adjusted $R^2$, Eqs. 6-8).

$$R^2_{adj} = 1 - (1 - R^2) \cdot \frac{n-1}{n-p-1} \tag{6}$$

$$MAE = \frac{\sum_{i=1}^{n} |Y_{p_i} - X_{obs_i}|}{n} \tag{7}$$

$$RMSE = \sqrt{\frac{1}{n} \sum_{i=1}^{n} (Y_{p_i} - X_{obs_i})^2} \tag{8}$$

To evaluate the relationship between $M_{obs}$ and $M_{sim}$, a linear regression was used (Eq. 9).

$$M_{obs} = \beta_0 + \beta_1 \cdot M_{sim} + \varepsilon \tag{9}$$

$R^2_{adj}$ (Eq. 6) was calculated to evaluate the explanatory power of the simulations while accounting for the risk of overfitting,
based on the total number of observations in the dataset ($n$) and the number of predictors included in the linear model ($p$),
including the intercept. In this context, p refers to the $M_{sim}$ derived from the scenario being evaluated.

$R^2_{adj}$ was selected as the *primary* ranking metric because it balances model fit with complexity, helping to identify the two
best simulation scenarios among the 105 scenarios that explain a high proportion of variance without overfitting the observed
data. Furthermore, $MAE$ and $RMSE$ (Eqs. 7-8) were used to assess the magnitude of prediction errors. $MAE$ provides a meas-
ure of the average magnitude of errors, while $RMSE$ emphasizes larger errors. In $MAE$ and $RMSE$, $Y_{p_i}$ and $X_{obs_i}$ represent the
simulated value of tree mortality rates from the scenarios being evaluated and the observed values of the tree mortality rates,
respectively. MAE was selected as the *secondary* ranking metric for the selection of the two top-ranked simulation scenarios.

Ultimately, to assess how spatial heterogeneity influences model performance, we fitted a generalized additive model (GAM)
relating Adjusted R² to the degree of local soil heterogeneity (σ).

All analyses were performed in R (v4.2, R Core Team, 2023) using the packages *data.table*, *tidyverse, mcg* and *rforclim*
(Dowle et al., 2019; Hiltner and Marano, 2024; Wickham et al., 2019).



# 3 Results

## 3.1 European beech

The relationship between soil water properties, simulated mortality, and observed mortality for *Fagus sylvatica* revealed clear patterns in model performance across the gradients of mean and minimum AWC (Figure 3A). In the lower-left region of the matrix—characterized by low mean and minimum AWC— there was poor agreement between simulated and observed mortality rates. Under these conditions, limited soil water buffering capacity accelerated the onset of drought stress, leading to abrupt and severe mortality events that do not align with observations. Overall, such scenarios resulted in early and intense

mortality signals prior to the year 2010 (cf. Figure SM 3.2.1.2). The high MAE (cf. SM 3.3.1) under low $AWC_{mean}$ (10-15 cm) underscored the model's overestimation of mortality, despite correctly identifying heightened vulnerability to drought.

In contrast, the regions of the parameter space with moderate to high mean and minimum AWC (i.e., towards the upper-right portion of the matrix) showed good agreement between simulated and observed mortality rates. Here, the model was able to capture both the timing and the relative magnitude of mortality signals. The corresponding lower MAE suggested that scenarios

with high AWC were less prone to overestimating mortality, i.e. these are conditions where AWC is sufficiently high to delay and dampen drought impacts. This pattern highlights the importance of mesic soils for moderating mortality responses, thus enhancing forest resistance to drought.

The transition from poor to strong model-data agreement (Fig. 3A) underscored the interaction between mean and minimum AWC in shaping mortality outcomes: when both were low, insufficient soil moisture buffering exacerbated mortality and

produced significant model–observation mismatches. Conversely, high $AWC_{mean}$ (e.g., $AWC_{mean} > 20$ cm) combined with high $AWC_{min}$ reduced the mortality response, improving model performance in terms of both trend (high adjusted $R^2$) and magnitude (low MAE).

Increasing soil heterogeneity (expressed as the difference between $AWC_{mean}$ and $AWC_{min}$) was associated with a clear shift in the mortality response (Figure 3B). Under homogeneous soil moisture conditions (i.e., soil heterogeneity class *None*), model

performance was poor (as judged by the median of the distribution), with simulated mortality exhibiting lower variability and very low agreement with observations. As a result, the lack of soil heterogeneity exacerbated extreme drought responses and increased the likelihood of mortality events. As soil heterogeneity increased, (i.e. *Low* and *Medium* classes in Figure 3B), the agreement between simulated and observed mortality rates increased as well, with a corresponding higher variability, particularly under the *Medium* heterogeneity class. This suggested that the spatial variability in soil water availability began to exac-

erbate mortality responses by creating localized zones of drought stress. At the highest levels of soil heterogeneity (i.e., *High* heterogeneity class), the adjusted $R^2$ was lowest, suggesting that abundant moisture across microsites prevented drought stress from fully developing, thus preventing drought-induced mortality to manifest in the simulated mortality rates. These overall patterns were confirmed by the analysis of the spatial heterogeneity via σ (cf. SM 2.2, Figure SM 2.2.1), where the GAM model suggested that the relationship between soil heterogeneity and model accuracy was highly nonlinear (cf. Table S 2.2).

Indeed, the adjusted $R^2$ initially increased as heterogeneity increased, reaching a peak at σ ≈ 0.34, after which model





performance declined sharply (Figure SM 2.2.1). Moreover, 47.8% of the variation in adjusted R² values across the dataset was explained by heterogeneity alone. This indicated a moderate to strong influence of local soil heterogeneity on model performance for European beech, suggesting that moderate levels of soil heterogeneity improved predictability, whereas high levels of heterogeneity reduced it.





**Figure 3.** Evaluation of the performance of soil moisture scenarios in reproducing simulated tree mortality compared to observed tree mortality for European beech-dominated sites. (**A**) AWC scenarios showing adjusted $R^2$ and MAE. The two top-ranked AWC scenarios are indicated by dashed red boxes. (**B**) Adjusted $R^2$ and simulated mortality rate across heterogeneity classes (cf. Fig. 2). (**C**) Simulated (the two top-ranked AWC scenarios) and observed mortality rates over time across all sites, and (**D**) model statistics for the two top-ranked AWC scenarios.



The observed drought-induced mortality rates of beech (Figure 3C) featured a narrow range and low interannual variability for 2000 to 2017, whereas the simulated rates from the two top-ranked AWC scenarios exhibited more pronounced fluctuations, indicating an overestimation of both magnitude and variability by ForClim (cf. Tab. SM 3.3.1 for complete statistics). Between 2000 and 2017, the observed annual drought-related mortality rate averaged 0.054% (±0.060%), whereas the two top-ranked simulation scenarios (84 and 90) produced substantially higher means of 0.228% (±0.175%) and 0.311% (±0.184%), respectively—approximately four times the observed rate.

Beginning in 2018, the observed mortality increased to 0.340% (±0.105%), reflecting the onset of the pronounced drought. Over this period, scenarios 84 and 90 projected mean mortality rates of 1.89% (±0.274%) and 2.05% (±0.304%), exceeding the observations by factors of about 5.6 and 6.0, respectively. Overall, during the 2018-2022 drought period, both observed and simulated mortality rates increase sharply, reflecting the expected impact of drought on forest mortality.

The statistical comparison between simulated and observed mortality rates (Fig. 3D) indicated that both two top-ranked AWC scenarios featured strong correlations with observed mortality, with $R_{adj}^2 = 0.73$. Despite this strong correlation, the MAE highlighted the model's tendency to overestimate the magnitude, with scenario 84 exhibiting a slightly lower MAE (0.48%) than scenario 90 (0.57%). The spread of the simulated values further emphasized the consistent overestimation of mortality in both scenarios, as the majority of the simulated values were above the 1:1 line. Overall, while the model successfully reproduced the temporal dynamics of drought-induced mortality of European beech, it systematically overestimated its magnitude.

### 3.2 Norway spruce

Simulation results for *Picea abies* revealed a complex interplay between soil water availability, soil water heterogeneity and drought-related mortality (Figure 4A). At very low $AWC_{mean}$ amd $AWC_{min}$, values—i.e., in the lower-left region of the matrix—model performance was relatively high when judget by the adjusted $R^2$. However, the mean absolute error (MAE) in this region of the parameter space reached 7.5–10%, indicating a notable mismatch between simulated and observed magnitudes of mortality. This pronounced mismatch indicates that the model responded too strongly to drought intensity. As mean and minimum AWC increased (i.e., towards the top right of the matrix, Figure 3A), model performance decreased, although the reduction in MAE suggests that discrepancies between simulated and observed mortality rates were reduced, particularly in the central and upper-right regions of the matrix (2.5% < MAE < 5%). Towards the upper part of the matrix—though not fully extending to the top right—both $R^2$ and MAE increased, highlighting a zone where the model better captures both spatial patterns and magnitudes of drought responses. Overall, rather than indicating a monotonic gradient of simulated drought resistance, the agreement across the matrix revealed a more complex pattern compared to European beech. For low $AWC_{mean}$, moderate adjusted $R^2$ and high MAE indicated that the model captured the general mortality pattern but exhibited high inaccuracies regarding the simulated magnitude. This suggests that low $AWC_{mean}$ was insufficient to buffer drought events and resulted in amplifying the mortality responses even when $AWC_{min}$ was relatively high. Yet, high $AWC_{mean}$ values buffered mortality events too strongly, leading to an underrepresentation of the recent drought-related mortality in the simulations.





**Figure 4.** Evaluation of the performance of each soil water scenario in reproducing simulated tree mortality compared to observed tree mortality for Norway spruce dominated sites. (**A**) AWC scenarios showing adjusted $R^2$ and MAE. The two top-ranked AWC scenarios are indicated by dashed red boxes. (**B**) Adjusted $R^2$ and simulated mortality rate across soil heterogeneity classes (cf. Fig. 2). (**C**) Simulated (the two best AWC scenarios) and observed mortality rates (dashed line) over time for Norway spruce, and (**D**) model statistics for the two best AWC scenarios.



The relationship between soil heterogeneity and model performance revealed a distinct negative trend for Norway spruce
(Figure. 4B). The adjusted R² values declined progressively from low to high soil heterogeneity, suggesting that the model
performed best in spatially uniform conditions and struggled to capture mortality dynamics in heterogeneous soils. This pattern
contrasts with *Fagus sylvatica*, where a hump-shaped response was evident (cf. Figure. 3B).

The GAM model confirmed that soil moisture heterogeneity had a significant non-linear impact on model performance, even
higher than in the case of European beech, with peak performance at intermediate levels of heterogeneity (σ ≈ 0.34), followed
by a steep decline in R² at higher σ (cf. SM 2.2, Figure SM 2.2.2). Notably, the decline in model fit at high σ was steeper for
Norway spruce, indicating a stronger sensitivity to higher soil moisture, whereas the performance of European beech decreased
more gradually. The intercept (0.113, cf. Table SM 2.2) suggested as well that when structural heterogeneity was minimal
(i.e., σ is close to zero), the model explained only about 11.3% of the variation in adjusted R² values — a lower baseline
compared to European beech, indicating that spruce model performance depended more strongly on soil heterogeneity.

Observed mortality rates of *Picea abies* between 2000 and 2018 were relatively low (mean = 0.526%, SD = 0.384%; Figure
4C); simulated mortality under the two top-ranked soil moisture scenarios featured comparable magnitudes (mean = 0.607%
and 0.555% for scenarios 76 and 90, respectively) but a somewhat larger variability (SD = 0.693% and 0.651%). This indicated
that ForClim captured both the level and the interannual variability of mortality in non-drought years well.

From 2019 to 2022, the observed mortality increased sharply (mean = 7.885%, SD = 2.560%), reflecting the impact of the
recent severe drought. Although both simulation scenarios captured the general trend of rising mortality during the early years
of this drought, with simulated means of 5.95% in both cases (SD = 5.50% and 5.29% for scenarios 76 and 90, respectively),
the model underestimated the magnitude of mortality in the later years, particularly in 2021 and 2022.

Between 2019 and 2022, the observed cumulative mortality reached 31.5%, with a pronounced peak of 9.6% in 2020. The two
simulation scenarios (76 and 90) featured a peak in 2019 and underestimated the overall severity, projecting a total mortality
of 23.8% during that period. Specifically, simulated mortality in 2020 was 7.8% under scenario 76 and 7.1% under scenario
90. Lastly, across 2021 and 2022, the observed mortality averaged 9.1%, whereas scenario 76 and scenario 90 yielded averages
of 1.6% and 1.9%, respectively. Overall, while the model reproduced the broad scale of drought-driven mortality, it did not
fully reflect the lagged mortality events after 2020. These findings were confirmed by the comparison in Figure 4D, where
simulated estimates aligned reasonably well with observations at low mortality rates but underestimated severity as mortality
rose, indicated by most of the high-mortality points lying below the 1:1 line for the last simulated years.

The bark beetle module induced a significant increase in mortality during the period of elevated drought, i.e. after 2018 (Fig.
4C), whereas in the absence of the bark beetle submodel, the overall trend and magnitude showed a substantial mismatch
compared to the observations (cf. SM 3.3.3, Figure SM 3.3.4). Thus, integrating the bark beetle submodel was essential for
capturing the drought-induced mortality wave of Norway spruce in German forests in the period 2018-2022.



## 4 Discussion

### 4.1 Trend and magnitude of simulated mortality rate

Our study demonstrated that a simple framework that semi-mechanistically considers the factors underlying drought-related tree mortality (i.e., the PIC scheme of ForClim v4.1 including the bark beetle model in ForClim v4.2) was able to capture the recent drought-induced tree mortality trends and particularly the response to the extreme drought of 2018-2022 at large spatial scales for two major European tree species, although model performance varied among species and revealed some limitations. Importantly, a key feature of the PIC scheme was its capacity to accurately predict the *timing* of extreme drought-related mortality events, although the *magnitude* of this mortality was captured differently for the two species, as discussed below.

For European beech, the model captured the pattern of mortality over time quite well, but it overestimated the magnitude of mortality throughout by a factor of 5 to 6, including the recent drought wave (2018-2022). This indicated that for beech the sensitivity to drought-related stress as parameterized in the PI scheme (Marano et al. 2025) is probably too strong. Yet, it is noteworthy that the *timing* of the drought-related mortality peak (2018-2022) was captured well; in the model, the *magnitude* of drought-related mortality depends on a single parameter whose value is going back to an assumption by Botkin et al. (1972), according to which stressed trees have just a 1% chance of surviving 10 stress years, thus yielding an annual stress-related mortality probability of 36.8%. Evidently, this probability was too high to be compatible with observed data for beech (George et al., 2022; Knapp et al., 2024). However, we wanted to assess whether the PI scheme was capable of predicting the timing of strong drought-related mortality (irrespective of its exact magnitude), and this was successful. Parameter calibration could certainly improve the fit between simulated and observed mortality rates, but this was not the scope of our analysis.

For European beech, our study is the first to explicitly simulate drought-induced tree mortality using a process-based dynamic vegetation model along a wide geographical gradient. The simulated mortality rates aligned closely in terms of *magnitude* with results from other studies examining drought-induced mortality during the same period in managed beech dominated forests, ranging between 0.5-2% at the patch level as well as the rising *trend* in mortality post-2018 (Meyer et al. 2022, Frey et al. 2022). This consistency across studies lends credibility to the model's capacity to replicate the pattern of drought-related mortality of beech, despite challenges in the direct comparison with the observations from a sample inventory such as WZE (see section 4.3).

For Norway spruce, the model captured the *magnitude* of mortality over time even better than for European beech, including the severity of the recent drought-induced mortality wave. Although some inconsistencies remained (e.g., minor temporal mismatches and overestimation of low-intensity events and the misaligned peak post-2018), these differences likely reflect the absence of model calibration and possibly also site-specific effects that the model cannot capture. In particular, the misaligned peak of bark beetle-related mortality might be attributed to uncertainty in both the model and the data. In ForClim v.4.2, the high sensitivity of Norway spruce to drought-induced stress overestimated the mortality response in the model for 2019, whereas the simplified bark beetle module may have underestimated the number of beetle generations, thus leading to the underestimation of mortality in 2021-2022. Empirical evidence also indicates that bark beetle outbreaks often lag behind




drought events by several seasons or years, as drought-weakened trees become increasingly susceptible over time (Seidl et al., 2008). Furthermore, in reality such outbreaks are typically tied to other disturbances such as windthrows, which the model

does not currently consider (Hlásny et al., 2021a; Jakoby et al., 2019; Netherer et al., 2019; Seidl and Rammer, 2017). These simplifications are likely to contribute to the mismatched peaks of simulated spruce mortality, underscoring the need for a more nuanced integration of delayed and compounding disturbance interactions.

Nevertheless, model performance was indeed better when bark beetle activity as a contributing factor to mortality was included; this underscores the importance of including biotic stressors in the PIC framework (Fischer et al., 2025; Trugman et

al., 2021). Our study demonstrated that the model's simplified PIC scheme successfully captured observed mortality rates, a noteworthy achievement given that previous statistical and process-based models that incorporated mechanistic approaches (e.g. hydraulic failure, carbon starvation) were largely unable to simulate drought-induced mortality (Petite-Cailleux et al. 2021, Yao et al, 2021, Davi & Cailleret, 2017, Fischer et al., 2025).

For Norway spruce, our modelling study outperformed precedent modelling attempts such as those reported by Anders et al.

(2025) and Fischer et al. (2025) as, despite no calibration, ForClim v.4.2 well reproduced the *magnitude* of the drought-induced mortality events and associated peak after the 2018 drought.

In comparison with recent modeling efforts (Anders et al., 2025; George et al., 2022, Fischer et al. 2025), ForClim v.4.2 accurately reproduced both the overall *trend* and *magnitude*, in case of European beech, and *magnitude* for Norway spruce of drought-related tree mortality. Thus, despite the inherent complexities of modeling drought-related tree mortality, our study

represents an important advancement by successfully addressing this challenge without resorting to calibration. In contrast to recent studies that used statistical modeling and thus large calibration efforts, which aligned closely with observed mortality data from the ICP monitoring network (e.g., Anders et al., 2025; Knapp et al., 2024), our primary objective was not to optimize the model for immediate predictive accuracy. Rather, we sought to explore and understand the processes underlying drought-induced mortality across different species and contexts, which is often impossible based on statistical methods alone (Brunner

et al., 2021; McDowell et al., 2013; Trugman et al., 2021). Our process-based approach thus prioritizes mechanistic insights over accuracy, allowing for the identification of key drivers of mortality. Future research should build on this foundation by structurally refining and then calibrating the PIC scheme.

## 4.2 Role of soil water availability and its local spatial heterogeneity for modulating species response to drought

To date, the quantitative importance of AWC for buffering or amplifying drought stress signals and tree mortality has remained

uncertain, despite its recognized relevance for modulating tree responses to water scarcity (Klesse et al., 2022; Martinez del Castillo et al., 2022; Mellert et al., 2018). Our findings provide detailed insights regarding the responses of *Fagus sylvatica* and *Picea abies* to varying degrees of soil water stress.

The overestimated mortality rates in the European beech simulations reported in our study are in stark contrast to the good match in the case of six mesic beech sites in Switzerland (Marano et al., 2025), and such difference can be attributed to several

reasons. First, it may be due to an inherent limitation in the model's assumptions for European beech under low precipitation





conditions. ForClim simulates European beech dominance within a relatively small optimum climatic space (cf. Bugmann, 1996; Bugmann and Cramer, 1998; Bugmann and Solomon, 2000), and thus may predict excessive mortality at sites with less than ≈700 mm of annual precipitation, as often found in Germany. In reality, populations of *Fagus sylvatica* may still thrive in such environments, likely due to local adaptations, microclimatic refugia, or access to groundwater—factors that lie outside

the current model structure. Consequently, the observed die-off in the simulations indicates that moisture availability, not just soil water-holding capacity, constitutes a critical bottleneck for accurately forecasting beech dynamics in marginally dry climates.

Moreover, the future of *Fagus sylvatica* is intricately linked not only to climate, but also to soil conditions (cf. Gessler et al., 2022; Meusburger et al., 2022; Walthert et al., 2021b; Walthert and Meier, 2017). Our study reinforced this notion, demon-

strating that soil properties and particularly their small-scale spatial variability play a critical role in modulating drought-induced mortality. The characteristics of the top-ranked scenarios for both species underscore the importance of moderate local heterogeneity of soil conditions with values of $\sigma = 0.47$ and $0.37$ for scenarios 84 and 90, respectively, for European beech, and $\sigma = 0.51$ for Norway spruce.

We found that *Picea abies* exhibits an even higher sensitivity to low AWC than *Fagus sylvatica*, which is in line with previous

research (Griesbauer et al., 2021; Henne et al., 2011; Lagergren and Lindroth, 2002). This may be linked to its shallow rooting system, which limits access to deep soil water during prolonged drought. Generally, higher AWC provides a critical buffer, allowing spruce to better withstand drought episodes and delaying the onset of mortality, which was confirmed by our simulations showing reduced model performance at higher AWC values. Specifically, also for Norway spruce the most predictive scenarios correspond to moderate levels of the heterogeneity of soil water availability ($\sigma = 0.51$ and $0.37$ for scenarios 76 and

90, respectively), suggesting that variability at the microsite scale (i.e., at the patch-level within the simulated forest stands) captures ecologically meaningful differences in moisture supply that influence tree vulnerability to drought.

Notably, scenarios characterized by either very low or very high heterogeneity (i.e., $\sigma$) performed less well for European beech, while for Norway spruce some low heterogeneity scenarios still performed well, although the discrepancies in simulated mortality were much higher compared to scenarios of medium heterogeneity. This indicates that for Norway spruce no variability

in soil moisture heterogeneity can heighten the mortality signal driven by drought (e.g., during the most recent massive drought), whereas excessive heterogeneity may introduce noise that masks the underlying drivers of mortality and dampens the signal. Thus, it is pivotal to account for the small-scale spatial variability of soil water properties when aiming to provide accurate mortality predictions in forest models based on semi-mechanistic (rather than merely statistical) frameworks. Overall, our research emphasizes that the future of beech and to some extent also Norway spruce hinges strongly upon soil conditions,

and general statements based on Species Distribution Models, which largely ignore soil properties, are likely to lack accuracy (e.g., Gessler et al. 2024).

Lastly, when attempting to use our PIC scheme for projecting the future fate of European beech and Norway spruce under climate change, we recommend to first perform a species-specific calibration. Furthermore, as climate change is likely to



exacerbate drought conditions across Europe, integrating detailed information on soil properties and species-specific drought resilience in forest models will be essential for accurately predicting mortality dynamics and, ultimately, guiding adaptive management practices.

## 4.3 Methodological considerations and research recommendations

Accurately predicting drought-induced tree mortality remains one of the most pressing challenges in forest modelling (cf. Fischer et al., 2025), hindered by both observational and methodological constraints.

First, long-term observations of drought-induced tree mortality remain limited, posing significant challenges for understanding, predicting, and comparing simulation results with observed mortality rates and trends (George et al., 2022). Such datasets are rare due to the difficulty of establishing and maintaining monitoring systems over extended, multi-decadal periods with at least an annual resolution, which however is essential for capturing drought-related mortality (Bugmann et al., 2019). This scarcity inherently limits the power of validation efforts for simulation models. We used data from ICP Level I plots, which provide a valuable compromise between spatial extent and temporal resolution, as they were surveyed systematically over decades with an annual resolution, providing a long-term standardized dataset where the causes of tree death are recorded. However, this dataset has limitations. The observations relied on just 24 trees per plot, making stand-level generalization impossible. Additionally, the data were not representative of true stand-level mortality dynamics, as they were collected on a per-tree basis rather than being representative of the stand. To address these limitations, it would be ideal if future monitoring efforts could focus on long-term, stand-level observational data to provide a comprehensive basis for validating and refining forest simulation models. Obviously, there are logistic constraints to this; the high-intensity Level II sites were set up to accomplish this, but they are lacking high replication, amounting to "just" 68 plots in total. In any case, the present study in the general context of understanding and predicting drought-induced mortality underlines the high value and necessity of long-term, high-quality monitoring data.

Second, the absence of stand-level observations necessitated the use of a simulation approach based on single species to provide the initialization dataset for the actual simulation study. This was a pragmatic and heuristically useful solution; it ignored the potential role of current forest structure and composition in mediating drought responses. Therefore, incorporating real forest states rather than initializing simulations from a "spin-up" is a critical step for better assessing model fidelity. Remote sensing technologies such as LiDAR combined with high-resolution aerial photography may be a promising approach to address the lack of ground observations and provide a first estimate of mortality rates after drought events (Mosig et al., 2024; Schiefer et al., 2023, 2024).

Third, these observational constraints are further compounded by the inherent complexity and uncertain relative importance of drought-related ecophysiological processes—including hydraulic failure, carbon starvation, and various abiotic and biotic stressors—along with challenges in determining the appropriate spatio-temporal resolution and level of model complexity needed for their accurate representation (McDowell, 2011; Sala et al., 2010; Trugman et al., 2021). Multiple studies have shown that models often fail to capture the interplay between predisposing factors (e.g., drought memory), inciting events



(e.g., prolonged water deficits) and contributing factors (e.g., pest outbreaks) (Fischer et al., 2025; Gazol and Camarero, 2022; Hartmann et al., 2018; McDowell et al., 2013). Trade-offs between model complexity, process knowledge and computational feasibility constitute further barriers to integrating physiological, ecological, and climatic processes at the scales needed to simulate large-scale tree mortality (Meir et al., 2015). These limitations underscore the need for an improved understanding of the processes that are triggering drought responses at the tree level and how they propagate at the stand and landscape scale (Choat et al., 2018; Hendrik and Maxime, 2017; McDowell et al., 2008). Furthermore, the development of hybrid modeling approaches that blend process-based and data-driven methodologies could be explored (Fischer et al., 2025; Zhang et al., 2021).

Lastly, recent studies (Anders et al., 2025; Fischer et al., 2025; Liu et al., 2021) have shown that model calibration offers a powerful means to align simulations with observations, enhancing model accuracy and robustness. However, to fully profit from calibration, it is crucial that models first demonstrate the capacity to reliably reproduce mortality patterns through process-based understanding. This ensures that the mechanisms driving mortality are captured accurately, rather than relying on statistical adjustments to match observed patterns. Calibrating models without a robust process-based foundation risks "getting the right answer for the wrong reason", where the output aligns with historical data but fails to reflect the underlying ecological dynamics, particularly since mortality models – including our PIC scheme – contain multiple parameters that are poorly constrained by direct observation, thus essentially constituting degrees of freedom for calibration, and thus entailing the risk of overfitting. Premature calibration would not only undermine confidence in the model but also jeopardize its ability to project future trajectories. By prioritizing process understanding, as done here, models are better equipped to capture the complexity of mortality drivers, enabling them to provide meaningful insights under novel and uncertain future conditions.

## 5 Conclusions

Our analysis evaluated the predisposing and inciting factor scheme (PI in ForClim v4.1) outside its original development context—Swiss beech-dominated forests—by applying it to broader datasets in Germany for two dominant species, European beech and Norway spruce. The scheme was further developed to account for contributing factors (PIC, giving rise to ForClim v4.2) in order to capture drought-related mortality signals across a wide climatic and pedospheric gradient for Norway spruce. From this research, we draw the following conclusions.

First, the model demonstrated its ability to replicate drought-induced mortality patterns with varying, but generally impressive degrees of agreement when compared to observed mortality rates. Discrepancies between simulated and observed mortality rates stem from several factors, including data imbalance (e.g., limited observations of dead trees), the high sensitivity of the bark beetle submodel to drought intensity, and the absence of model calibration to the specific conditions of the analyzed datasets.

Second, soil water availability emerged as a key factor influencing species-specific responses to drought, both in terms of absolute AWC (available water holding capacity) values and the local spatial heterogeneity of soil conditions modeled through



555 varying AWC scenarios. Low AWC conditions accelerated and amplified mortality, whereas moderate to high AWC buffered drought impacts, improving alignment between simulated and observed mortality trends. Low soil heterogeneity correlated with increased mortality events, as the lack of microsite variation provides limited opportunities for trees to access favorable soil conditions under drought stress, whereas higher soil heterogeneity mitigated drought impacts as it provided local refugia of higher water availability. These findings underscore the critical role of soil water dynamics in shaping tree resistance to drought, in addition to mere climatic factors.

560 Third, by incorporating a bark beetle module, ForClim v4.2 achieved greater accuracy in simulating Norway spruce mortality patterns under drought stress compared to simulations that did not consider bark beetle activity.

Lastly, further research is needed to enhance the robustness and applicability of the PIC scheme, regardless of the degree of ecophysiological detail being employed. Specifically, higher-resolution data, such as annual observations of tree mortality and crown condition at the stand (rather than single-tree) scale, would be important for refining model predictions. Moreover, 565 model calibration could be used once the primary processes driving drought-related mortality are better isolated and understood. Such advancements hold the potential to not only improve the accuracy of process-based models under historical climate, but most importantly to offer deeper insights for disentangling the complex, multifactorial drivers of drought as a compound phenomenon. This, in turn, would pave the way for more reliably simulating the impacts of future climate change, thereby supporting adaptive forest management strategies in an era of intensifying climatic stressors.

570 **6 Supplementary materials**

The following supplementary materials are attached to this article:
- *Supplementary Material 1*: Additional figures and tables.
- *Supplementary Material 2*: ForClim model documentation.

**7 Code and data availability**

575 The ForClim v.4.2 model source code, simulation settings and outputs are archived at the following Zenodo (Marano and Bugmann, 2025). Analysis code in R scripting language is available are also archived at Zenodo (Marano and Bugmann, 2025).

**8 Author contribution**

Gina Marano, Harald Bugmann and Ulrike Hiltner conceived the idea, developed the concept and framing of the paper; Gina Marano curated the ForClim source code, conducted the analyses, created the graphics, and drafted the main manuscript; 580 Nikolai Knapp provided the ICP-Level 1 data. All authors contributed to the writing, reviewed the manuscript and approved its submission.



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
