# Peer review of "Simulating the recent drought-induced mortality of European beech (Fagus sylvatica L.) and Norway spruce (Picea abies L.) in German forests"

_EGUsphere, 2025_

## Author Comment (AC2)

**Author responses to Reviewer 1 of egusphere-2025-1534**

We sincerely thank Reviewer 1 for the thorough and constructive evaluation of our manuscript. We greatly appreciate the valuable comments and suggestions, which have helped us improve the quality and clarity of the paper. Our detailed responses to each comment are provided in red font below. All revisions made to the main manuscript are indicated by line numbers and refer to the new revised version. Additional analyses, figures, and tables have been incorporated into the revised manuscript and Supplementary Material (hereafter SM) as appropriate.

We substantially revised and extended the manuscript. Specifically, we (i) clarified the role of "influential outliers" and added new analyses quantifying model performance for the pre-2018 period (Results §3.1 and 3.2, SM 3.3.1); (ii) deepened the mechanistic interpretation of the post-2017 mortality surge by analyzing the relative importance of predisposing, inciting, and contributing factors (Results §3.1 and 3.2, new Fig. S 3.3.2, and new Discussion §4.3); (iii) conducted a targeted sensitivity analysis for Norway spruce key parameters (kDrSc and Pbark) to assess model robustness (new SM §3.3.7, Table S 3.3.7); (iv) examined spatial patterns of observed and simulated mortality using spatial autocorrelation metrics (new SM §3.3.3, new Figs. S 3.3.3.1–S 3.3.3.2); and (v) improved figure readability (Figures 2,3,4), clarified methodological descriptions (soil parameterization, model setup), and corrected minor errors and typos throughout.

We believe these changes strengthen the manuscript's transparency, reproducibility, and mechanistic interpretation of drought-induced tree mortality.

**RC1 comments**

In this study, the authors present a refined version of the DVM ForClim (version 4.2) with the aim to more accurately simulate drought-induced mortality of Norway spruce and European beech. In particular, they implement predisposing, inciting, and — in the case of spruce - contributing factors, which they termed a PI(C)-scheme. Importantly, the authors do not per se calibrate their model against observations in order to test whether their model implementation mechanistically captures mortality.

The presented results indicate, that for both species the stark increase in mortality observed during and after the extreme 2018 drought is reproduced. Yet, absolute mortality rates were largely overestimated for beech whereas the ongoing high mortality of spruce was not captured by the simulations. Based on 105 AWC-simulations, the authors moreover conclude a high importance of soil properties and soil heterogeneity (particularly for beech). Finally, for spruce only the simulations including a bark-beetle component were able to reproduce recent mortality rates. Eventually, the authors advocate for incorporating such mechanistic schemes into DVMs rather than striving for statistical/empirical models while also stressing the importance of an actual calibration of the model if simulating mortality under future conditions. As such, the study touches an important topic in context of dynamic vegetation models, namely the incorporation of drought-induced mortality which to date remains a major challenge. Consequently, the study can be considered very suitable to the general audience and scope of GMD.

Yet, before being publishable some major aspects have to be considered.

Firstly, while I particularly appreciate the approach not to calibrate their model against observations, I wonder to what degree the deployed 105 different AWC-scenarios in combination with the observed mortality rates (which feature a stark increase after 2017) do not result in similar problems arising from classic empirical models. In particular, I wonder to what degree the high error squares introduced by the high mortality rates after 2017 act as 'influential outliers' which have the potential to largely boost the main evaluation metric of r2. Given this, it seems likely that those AWC-models are selected which match best the observed mortality increase after 2017. But do they also represent the models with the best mechanistic mortality implementation? In other words: how would your models perform if only simulating the years 2000-2017? In particular for spruce this seems to play a role since – based on supplementary figure 3.2.2.2 – the peak of mortality in some simulations occurred in 2021 (instead of 2018) or sometimes even already in 2017. Since all of the 105 simulations are based on similar model parameters, I wonder how such different mortality peaks can be achieved (stochasticity?) and to what degree the mortality-implementation really can be considered robust. I believe these points needs to be clearly highlighted when interpreting the results, since I am guessing that the r2 will largely drop if not applying the model to the full period. At least, the authors should – in addition to results representing the full period – show how consistent their model evaluation is if excluding the years after 2017 to avoid the influence of these extreme years. This would then provide a better picture on how the mortality implementation performs under less dry conditions (for instance 2003 and 2015 were also pretty dry in Germany). And it would tackle my concern, that the model-selection procedure is biased by influential outliers, i.e. the extreme impact of the 2018 drought.

**Author response:** The reviewer raises important aspects here. Indeed, the R2 is boosted by the 'influential outliers', i.e. the mortality wave post-2017. When the performance pre-2017 is considered, the skill of the model is much lower. Yet, it has to be taken into account that observed mortality rates are very low in this period (for beech even more so than for spruce), which implies that the uncertainty in the observed data is large, among others due to the very small sample size. Furthermore, the simulated mortality in this period is mostly 'background' mortality, i.e. it cannot be attributed to any particular cause and thus should best be viewed as being stochastic. Thus, there is no expectation that the model would match the pattern of the pre-2017 mortality well, and we prefer not to add such analyses to the manuscript. Regarding the temporal placement of the peak of the mortality as a function of the AWC scenario, it is important to note that about 40% of the scenarios feature the peak in 2018 or 2019 for beech, and about 30% feature the peak in 2019 for spruce. Yet, we set up the 105 scenarios to cover a truly broad suite of AWC conditions, including scenarios that feature very low (i.e. unrealistically low) values, and it is these scenarios that give rise to erroneous mortality peaks e.g. in 2003 and in other years. We now explain this in the revised manuscript at lines 372-378 for beech and lines 447-453 for spruce in the Results section, as well as in the Discussion section (lines 488-495). Additionally, we have added a figure in the supplementary material (SM 3.3) "Mechanistic understanding of mortality: periods, PIC, spatial patterns", Figure S 3.3.1).

Secondly, while the authors conclude that incorporation of a bark-beetle component as well as soil properties (mainly AWC) appear as major drivers of tree mortality, the model-mechanisms causing the stark increase of mortality after 2017 are barely discussed. To provide the full picture, the authors should more deeply explore their model output in order to understand which environmental driver variables are responsible for the strong increase in mortality. Is this simply related to the extraordinary drought of 2018 or are there also predisposing factors (e.g. the dry year 2015) that contribute to this increase? This might also help to explain, why the observed ongoing high spruce mortality after 2018 is not captured by the DVM and it would also provide a better understanding of what may be simulated if applied to climate projections.

**Author response:** This is an important aspect indeed. In order to answer the reviewer's questions, we have added the analysis of the stand-level average (i.e., what fraction of the dead trees is experiencing the predisposing factors (slow-growth vs. drought-memory induced), as well as the inciting factor (fraction of trees affected). This is now shown in SM3, Figure S 3.3.2. We have also added an explanation on these results for beech (lines 379-387) and spruce (lines 468-479).

Thirdly, I understand why and generally agree with the authors not wanting to calibrate their model against observations. Yet, some of the model parameterizations appeared somewhat arbitrary to me and I wonder whether a sensitivity analysis of specific parameters wouldn't be meaningful to gain a better understanding of model behavior, which I think should be more emphasized in a model development framework. For instance, the classification of base annual probability for the bark-beetle outbreak classes as well as the factor of 2/3 applied for the inciting factors seem arbitrary but likely have an impact on the model outcome. For future implementations of the PIC-scheme it would be very helpful to know how sensitive the model reacts to these metrics. In other words: would it be possible to achieve models with similar or even higher performance if choosing different outbreak classes or a different factor? And to which of the two factors is the simulated mortality more sensitive? This information would provide readers with more guidance on how to implement comparable mortality mechanisms in other DVMs.

**Author response:** We appreciate the reviewer's suggestion and agree that targeted sensitivity analysis is important in a model development framework, especially when parameters are motivated by process understanding rather than being calibrated. We therefore have performed a local sensitivity analysis for spruce around the two best scenarios (i.e., no. 29 and 42) for the two parameters highlighted by the reviewer, i.e. the drought scaling factor (kDrSc) and the baseline outbreak probability ( $P_{bark}$ ).

Specifically, we have perturbed one factor at a time by  $\pm 10\%$  and  $\pm 20\%$  and summarized the fit using MAE (i.e., lower is better) and adjusted  $R^2$ . We quantified sensitivity as a central-difference slope (change in the fit metric per 1% parameter change, par) with 95% bootstrap confidence intervals obtained by resampling years. The simulation runs from the originally submitted manuscript served as the baseline. Across both scenarios, simulated mortality is more sensitive to kDrSc than to  $P_{bark}$ . Point estimates of  $|\frac{\partial MAE}{\partial}par|$  are  $\sim 10\times$  larger for kDrSc than  $P_{bark}$  in scenario 29 and materially larger in scenario 42 (cf. new Table S 3.3.7). Increasing kDrSc tends to reduce MAE and increase  $R_{adj}^2$ , while  $P_{bark}$  yields near-flat responses within a

change of  $\pm 20\%$ . Scenario 29 shows that kDrSc + 20% improves MAE by ~2.4% and increases  $R_{adj}^2$  by ~0.008. The response under scenario 42 is flatter: MAE changes are ~ $\pm 0.2$ –0.3% and small  $R_{adj}^2$  gains are possible at negligible MAE cost. We have added the above-described sensitivity analysis to the manuscript lines 287-295 (Methods), lines 459-467 (Results), and its supplement (cf. new SM Section 3.3.7 and Table S 3.3.7). Overall, these findings demonstrate that the model behaves robustly with respect to both parameters and that the specific parameter choices made in this study do not critically affect model performance or interpretation.

Finally, while the comparison of model output and observations is based on mean mortality across all sites, the spatial scale is largely ignored. I understand why this is the case (only few mortality observations and stochasticity of the model likely result in spatially varying patterns) but it nevertheless deserves a mention in the discussion and maybe 1-2 display items in the supplementary to visualize the spatial patterns of simulated mortality. For instance, I wonder whether simulated mortality shows a spatial pattern or a rather random structure. If the former (spatial patterns) this might also point at the environmental drivers being mostly responsible for the mortality increase (see my second point above).

**Author response:** We thank the reviewer for this comment. We decided to assess the spatial patterns of observed and simulated mortality separately, rather than only their differences (cf. SM section 3.3.3. Spatial mortality). Specifically, we quantified global and local spatial autocorrelation for both European beech and Norway spruce plots to detect spatial clustering or dispersion. This approach provided a quantitative and spatially explicit assessment, besides comparing observed and simulated maps (cf. panels A of Figures S 3.3.3.1 and S 3.3.3.2 respectively), and the results confirm that both datasets exhibit very weak or absent spatial structure, consistent with the reviewer's expectation. This consideration has been added to the main MS, lines 370-371 (beech) and line 446 (spruce).

Only if these additional aspects have been taken into consideration the manuscript will transparently show how the suggested PIC-scheme may enhance the accuracy of mortality simulations in DVMs which I believe should be the major aim of the study. And even if some of the currently very convincing model performance evaluations (r² of 0.72 for beech!) would drop under a corresponding reanalysis (e.g. if adding validation metrics representative of the period 2000-2017 only) this is valuable and important information to the readers, since it would reflect that matching extreme patterns not necessarily means that mortality is generally well implemented (i.e. under less dry conditions). This in turn would also indicate the necessity to only very carefully interpret model output if applied to future climate projections. And finally, understanding what exactly drives the enhanced mortality after 2017 within the model may shed more light on the actual mechanisms driving tree mortality although inference from mechanistic models should be undertaken carefully.

Please find more detailed comments referring to specific sections of the manuscript below.

Abstract

Line 16: isn't hypothesis 3 a logical consequence of 2, i.e. if soil properties have a strong influence, local soil heterogeneity will automatically have a modulating impact.

**Author response:** We have incorporated H3 into H2 as suggested (cf. lines 17-20).

Line 21: please quantify 'hundreds'. How many plots in total?

**Author response:** Done (line 23).

Introduction:

In contrast to the abstract, you here combined the second and third hypothesis from the abstract. I personally prefer this combination, since H2 and H3 in the abstract are closely related. I suggest to adapt the 3 hypotheses also in the abstract (see also my point above).

**Author response:** Done (cf. lines 17-20).

Methods:

Line 100: What is the reason for the large gap in northeastern Germany. Aren't there any sites with beech or spruce? I would at least expect a couple of beech sites here and there. If not, please briefly mention the reasons for this geographic gap.

**Author response:** We selected only sites with documented drought-induced tree mortality (line 111), similarly to Knapp et al. 2024 and we referred to the author's same mortality observations, which explains the absence of sites in northeastern Germany. We have clarified this in lines 111 and 122.

Line 147: It seems that Marano et al., 2025 is currently under review. This obviously hampers the inspection of details as suggested. Would it make sense to show these details in the supplementary?

**Author response:** As the reviewer requested, the main equations of Marano et al. (2025) have been included in the new "SM 4 | ForClim 4.1: Predisposing and Inciting factor scheme".

Line 150: If I understand correctly, the heterogeneity was artificially generated. I wonder, how this reflects actual soil heterogeneity. And it isn't fully clear to me, whether you actually used existing soilmaps to characterize the soil properties which eventually determine AWC (I later learned this information comes below). Since soil properties are quite crucial for drought-related mortality (as you claim yourself), the soil parameterization is a quite crucial step, which deserves a more detailed description.

**Author response:** We agree that soil characterization is a critical aspect of modeling drought-induced mortality. As described in the submitted manuscript (lines 226-231, old version), we initially attempted to derive site-specific soil water holding capacity (AWC) and its spatial heterogeneity from soil maps (BGR, Meinel 2015), but found these data to be ecologically implausible, as described in the original submission. Therefore, rather than relying on such products, we resorted to a scenario-based sensitivity analysis, spanning a plausible yet admittedly expert-based range of AWC values and heterogeneity levels. This approach allowed

us to systematically test how variations in soil water availability and heterogeneity modulate drought-induced mortality, acknowledging that this constitutes an exploration of plausible conditions rather than a reflection of true soil heterogeneity. We have clarified this rationale in Section 2.3.4 (lines 231-233).

Line 182: please reword: 'This heuristic approach we used combined'

Author response: Done, line 185.

Line 182: did you run a sensitivity analysis to see how these somewhat arbitrary boundaries affect your model outcome? Might be worth a try to see how influential this classification is and whether a different classification might provide better/different results.

**Author response:** We agree that the classification boundaries for bark beetle outbreak probabilities introduce a degree of arbitrariness. Yet, we used this classification just to derive the value of the parameter  $P_{bark}$ , based on averaging the outbreak map of Hlasny et al. (2021). As explained in our response to the general comments (above), we conducted a targeted sensitivity analysis showing that this parameter choice has only a minor influence on model performance. Thus, the specific classification scheme does not critically affect the model outcomes.

Line 189: how is the stress status of trees defined/quantified? Please elaborate.

Author response: The sentence was ill placed and has been deleted and reformulated (line 190). The explanation is following in the text just below. The term "stress status" used in the manuscript refers to the algorithmic check of whether a tree meets predefined environmental stress thresholds, in this case, drought stress indicated by the annual drought index (mDrAn). When this threshold is crossed, the tree is considered a candidate for the application of stress factors in the PI scheme, including potential bark beetle infestation.

Line 196: this factor (2/3) is again somewhat arbitrary and would require a sensitivity analysis to quantify its impact on the model outcome.

**Author response:** As per the reviewer's request, we have performed a targeted sensitivity analysis to quantify the impact of the 2/3 scaling factor (kDrSc) on model outcomes for Norway spruce. The results, summarized in Table S2, show that simulated spruce mortality is moderately sensitive to this parameter: increasing kDrSc by +20% improved model fit (MAE -2.4%,  $R_{\rm adj}^2+0.008$ ), while decreasing it by -20% had the opposite effect. In contrast, changes to the baseline bark beetle probability ( $P_{\rm bark}$ ) produced negligible effects within the same range. Overall, these results indicate that model behavior is primarily governed by drought-related processes, and that the current parameterization is robust, meaning that small variations in either factor do not materially alter model performance or interpretation. This is clarified in lines 459-467.

Equation (5): according to equation 3, gGen can only reach values between 0 and 1 or exactly 2. I wonder whether this abrupt jump from (less than) one generation to 2 generations isn't arbitrary or maybe if there is a typo in equation 3 based on the query of 1.5 here. In any case, the threshold of 1.5 generations is again somewhat arbitrary. Please verify and potentially elaborate.

**Author response:** We thank the reviewer for this helpful comment. Indeed, the linear term in Eq. (3) was missing a factor 2. This has now been corrected (line 193). The choice of the 1.5-generation threshold is not arbitrary but reflects observed elevational phenology patterns of *Ips typographus* in Switzerland as in Jakoby et al. (2019). In their work, the authors demonstrated a strong negative relationship between voltinism and elevation and reported that at approximately 1,000 m a.s.l. the mean number of generations is ~1.5, i.e. "in about 50 % of all years two, otherwise one generation" (their Fig. 3b and accompanying text). Our corrected equation yields  $g_{\text{Gen}} = 1.5$  at  $mDD_{An} = 1600$  degree-days, which corresponds closely to this elevational band under current Swiss climatic conditions. Consequently, the criterion  $g_{\text{Gen}} > 1.5$  identifies years or sites whose thermal regime consistently favors bivoltinism, a biologically important tipping point for mass-attack risk both in reality and also within our PIC framework. We have clarified this rationale in the Methods (lines 206–209) and now explicitly cite Jakoby et al. (2019) as the empirical basis for the 1.5-generation cutoff.

Line 219: that's the information I was expecting above. Maybe briefly mention above and refer to this section.

**Author response:** Done**

Line 236: I suggest to show a supplementary display item which depicts the original data and in comparison - the min and mean AWC values achieved by your approach to reflect how much of the original spatial variance in AWC is retained in your data. At current it is not clear to me how well your AWC-scenarios actually mirror reported AWC.

**Author response:** As explained on l. 226-227 of the submitted manuscript, the range of values for the 105 scenarios was taken from the mapped BGR product (cf. Figure S 2.1.1 in which the values are evident from the legend), whereas we did not use the spatial variance of the BGR product because we deemed it unreliable (as explained on l. 227-230 of the submitted manuscript, see also Figure S 2.1.2); we therefore do not think that comparing spatial patterns or the frequency distribution of the BGR data would be useful. Note that in our approach, at each ICP Level I plot the entire range of the scenarios is explored.

Line 239: what is the reason for choosing this specific period, i.e. 2000-2022?

**Author response:** We selected this period as it encompasses the most recent, well-documented drought events in Central Europe, notably the droughts of 2003, 2015, and 2018–2022, which have been extensively studied for their ecological impacts. Our aim was to assess model

performance against mortality patterns associated with these droughts. Starting in 2000 allowed us to capture pre-drought baseline conditions, enabling a comparison of forest dynamics before, during, and after these critical events, and particularly checking whether the new mortality scheme is capable of simulating the 2018ff. mortality wave while not simulating mortality in 2003 or 2015. An explanatory sentence has been added (lines 248-249).

Line 243: A subtraction can lead to negative mortality rates. Did you encounter this? If so, how did you treat this?

**Author response:** We did not encounter negative mortality rates. This is because the simulations are averaged across multiple plots, resulting in simulated background mortality values that are very close to the theoretical ones (0.921% and 0.495%). The subtraction is performed as a post-processing step (i.e., not within the simulation itself), which ensures that negative values do not occur.

Line 269: From equation 9 it seems you only used Msim, so why are you concerned about overfitting? Or did I miss something? Well, if you're concerned about overfitting, variance inflation should be considered e.g. by computing VIF for the predictor variables and excluding highly co-linear predictor variables (but again, if only using one predictor variable this does not make sense). So, I wonder which predictor variables you've been using at all. Please clarify and – if necessary - elaborate.

**Author response:** Indeed in our analysis we used only one predictor variable  $M_{sim}$  in the linear regression of Eq. (9), meaning overfitting due to multicollinearity or excess predictors is not an issue here. This was an error in the text, and the sentence has been corrected (line 280).

Line 275: I assume your R2 adj values do not follow a Gaussian distribution since ranging from 0 to 1. Did you account for this in your GAM? Which datatype/family did you specify in your GAM? binomial? Please elaborate.

**Author response:** We thank the reviewer for this insightful comment. Although adjusted R2 values are indeed bounded between 0 and 1, their distribution in our dataset was approximately symmetric and unimodal, without values close to the boundaries. Therefore, we fitted the GAMs using a Gaussian family with identity link, which is appropriate for continuous response variables that are roughly normally distributed within this range. Diagnostic checks confirmed that residuals were homoscedastic and approximately normally distributed. Consequently, a transformation or alternative family (e.g., binomial or beta) was not required. We have added the specification the GAM family that we used to the caption of Table S 2.2.

**Results:**

Fig. 3, panel A: the dark end of the color scale for R2 does not always allow for depicting the size of MAE. Please adjust. Same for Fig. 4

**Author response:** We have revised the color palette of all main figures, namely Figures 2, 3 and 4 to enhance readability and ensure all elements within each plot can be recognized easily.

Lines 325-336: I wonder to which degree the overall variance of the data affects your r². It would be interesting to compute r² for the period before 2018 only to see how well the 'average' mortality under less dry conditions is captured by the models. It seems, that your model parameterization is able to capture the stark increase in mortality after 2017 but I wonder to what degree the model mechanistically captures mortality or whether it simply reacts to one extreme year. This aspect deserves more careful thinking and interpretation, particularly if using the model later on to predict mortality rates based on projected climate data. This does not require to rerun the simulations but only to evaluate their performance for a sub-period which is a common procedure when evaluating model performance.

**Author response:** Following our first response to the General Comments, we have included an additional breakdown of this evaluation in the Supplementary Material (cf. SM Section 3.3.1) and in the main MS in the Results section for beech (lines 372-378) and for spruce (lines 447-453) as well as in the Discussion (cf. section 4.1, lines 488-495).

Fig. 4: to avoid misinterpretation I suggest to use the same range of  $r^2$  values in the legend as for beech to visually highlight that  $r^2$  is much lower for spruce.

**Author response:** Following the reviewer's recommendations, the scale in Figures 3 and 4 has been adjusted accordingly to improve the readability of the two figures.

Line 386: Again, I wonder to what degree the extreme years after 2017 affect your model-selection process. Moreover, while I agree that the bark-beetle model is important to incorporate, it yet seems to require some improvements, given the inability to capture prolonged impacts of the 2018 drought. Also, from Fig. 3.2.2.2 in the supplementary it seems that some model runs obtained quite different mortality peaks (some in 2017, some in 2021). Since – if I understood correctly – the only difference in these runs was the AWC implementation, I wonder which circumstances have driven such temporally inconsistent mortality peaks. As suggested above, I suggest to gain a deeper understanding of the actual climatic forces driving the mortality peaks, since this also would allow for a better mechanistic interpretation of the parameterization.

**Author response:** We agree that the post-2017 period, characterized by extreme drought events, has a strong influence on the model selection process and interpretation. To quantify this effect, we compared the explanatory power (R2) of the model calibrated for the full observation period versus the pre-2018 period (Fig. S 3.3.1A–C). For both species, model fits typically improved when including the post-2017 data ( $\Delta R^2 > 0$ ), indicating that the extreme years provide valuable information on system behavior under unprecedented stress conditions rather than distorting the model selection.

For spruce, the inclusion of 2018–2020 particularly improved the fit of simulations that included bark-beetle dynamics (Fig. S 3.3.1 D-F), suggesting that the beetle submodel captures

the abrupt mortality pulse triggered by the 2018 drought. However, we acknowledge that the model still underestimates the legacy effects of bark beetle (line 454), likely because it does not fully account for lagged physiological decline associated with secondary beetle outbreaks following the initial event.

Regarding the temporal inconsistency of simulated mortality peaks (2017 vs. 2021), this variation primarily stems from differences in the available water capacity (AWC) parameterization. Scenarios with lower AWC experience stronger water deficits (low drought index values), leading to earlier and sharper mortality peaks (often in 2003 or 2018). In contrast, higher AWC buffers drought impacts, delaying mortality peaks to later years (e.g., 2021). Thus, these differences are not random but reflect distinct drought sensitivities of the simulated stands, reinforcing that AWC is a critical control on drought-induced mortality timing and magnitude.

In response to the reviewer's suggestion, we have added an explicit reference to the role of drought intensity in both the Results for beech (lines 336-336 and lines 358-360) and for spruce (lines 420-422) and Discussion sections (lines 490-492) respectively and have provided a mechanistic interpretation of model behavior in the new Discussion section 4.3.

**Discussion:**

Line 446: but it is not yet fully clear what these key drivers are. in other words: which environmental circumstances have led to the stark increase in mortality after 2018? Please elaborate.

**Author response:** The environmental drivers in question are mainly the drought memory and the annual drought index, which are part of the predisposing (drought memory) and inciting (drought intensity) scheme. As noted above, these considerations have been added new discussion Section 4.3.

Line 464: I generally agree that soil properties are important in mediating drought but some care needs to be taken when interpreting model performance since what you describe here most likely relates to your model-specific parameterization of beech. In reality this small-scale variability might not be as important for a relatively anisohydric species with relatively deep rooting systems.

**Author response:** We agree that the influence of small-scale soil heterogeneity on drought-induced mortality, particularly for anisohydric and deep-rooting species like European beech, warrants careful interpretation. Yet, beech's capacity to buffer drought stress due to deep rooting is debated (e.g., Gessler et al. 2022). Moreover, in drought-prone environments and on soils with restricted rooting depth (due to bedrock, compaction, or poor structure), even beech may depend on variable water availability within the upper soil profile, thereby rendering microsite heterogeneity relevant (Walthert et al., 2021). Thus, we would like to maintain our argument.

Line 484: Again I do agree, that soil conditions are important but we have to keep in mind that your interpretation relies on model output and thus mirrors how the model was parameterized. This not need to directly mirror reality. Thus, I would be more careful when deriving implications for real systems from model output.

**Author response:** We agree that model-based interpretations must always be made with caution. Yet, our study relies neither on model calibration nor on empirical fitting, but on ecologically grounded representations of soil-plant interactions. We thus are convinced that the emerging patterns, particularly the role of soil water availability and heterogeneity in modulating drought-induced mortality, are robust, and they are also consistent with empirical evidence (e.g., Walthert et al., 2021). Upon re-reading of these portions of the Discussion, we feel that the statements are cautious enough, and we would like to maintain them.

Line 488: when doing a species-specific calibration, a robust cross-calibration verification should be undertaken to avoid artifacts introduced by influential outliers (as the years after 2017). Please elaborate.

**Author response:** With this text, we wished to imply that that species-specific calibration should be accompanied by a robust validation strategy, such as cross-validation or split-sample testing, to avoid artifacts caused by influential outlier years, particularly extreme drought periods like those after 2017. We reformulated the sentence and added a cautionary remark (lines 653-655).

Line 539: You stressed to prioritize process understanding. Yet, the processes leading to the increased mortality after 2018 are barely discussed. Is this mostly related to one extremely dry year (2018), ongoing soil-drought, or predisposing factors? Please evaluate your model output accordingly to provide a deeper understanding of the underlying mechanisms.

**Author response:** The reviewer raises a crucial point, which we now have discussed extensively in the new Discussion section "4.3 Mechanistic interpretation of the post-2017 mortality surges".

**References**

Gessler, A., Bächli, L., Rouholahnejad Freund, E., Treydte, K., Schaub, M., Haeni, M., ... & Meusburger, K. (2022). Drought reduces water uptake in beech from the drying topsoil, but no compensatory uptake occurs from deeper soil layers. New Phytologist, 233(1), 194-206.

Walthert, L., Ganthaler, A., Mayr, S., Saurer, M., Waldner, P., Walser, M., ... & Von Arx, G. (2021). From the comfort zone to crown dieback: sequence of physiological stress thresholds in mature European beech trees across progressive drought. *Science of the Total Environment*, 753, 141792.

---

## Author Comment (AC3)

**Author responses to Reviewer 2 of egusphere-2025-1534**

In response to Reviewer 2's valuable comments, we improved the clarity, consistency, and transparency of the manuscript and supplementary material (hereafter SM). Specifically, we (i) clarified the definitions, numbering, and selection criteria of the best-performing scenarios and corrected figure annotations accordingly (Figs. 3 and 4); (ii) explicitly referenced the comparison of ForClim 4.1 and 4.2 in the text (lines 454–458, Fig. S 3.3.6); (iii) performed and described a local sensitivity analysis of key parameters (kDrSc and Pbark) to quantify model robustness (new SM section 3.3.7); and (iv) harmonized equation order, figure legends, and numerical precision throughout the manuscript.

These adjustments enhance the manuscript's readability and ensure consistency between text, figures, and supplementary materials.

**RC2**

Marano and colleagues present a new version of a Dynamic Vegetation Model (DVM), namely ForClim. In the presented version 4.2 of ForClim (ForClim 4.2), the authors aim to improve the simulation of drought-induced mortality of Norway spruce and European beech. In particular, the authors implement a scheme accounting for predisposing, inciting, and contributing factors. The addition of these schemes results in a more accurate representation of the observed mortality during the 2018-2022 drought period. Although the manuscript is well-written and presents interesting results, some clarifications are needed before publication, as listed below.

**General Comments**

It would be informative to include a comparison of the same selected scenarios (76 and 90) with and without the bark beetle submodel within the main text. The addition of this comparison would provide the reader with a direct visualisation of the model enhancement gained in ForClim 4.2 compared to ForClim 4.1.

**Author response:** The two model versions comparisons, namely ForClim version 4.1 (no bark beetle module) and version 4.2 (with bark beetle module) were already contained in the submitted manuscript. In the revision however, we referred more explicitly to Figure S 3.3.6 in SM (lines 454-458, main manuscript). We decided to place the figure showing the simulations conducted with v.4.1 in the supplementary material, and not in the main MS, to avoid content overload.

Besides, the authors do not perform tuning of the used parameters since they want to emphasise the improvement provided by adding predisposing, inciting, and contributing factors within the model. However, it would be beneficial for the reader to know how much ForClim respond to changes in the new parameters.

**Author response:** We agree that further insights into model sensitivity to the new parameters would be useful. Note that in response to the requests by RC1, we have undertaken a local sensitivity analysis to substantiate our choice of parameter values. Results can be found in Table S 3.3.7 in the Supplementary Material 1.

Specific Comments

Eq 6-8: put in the same order as the introducing list: MAE, RMSE, and adjusted R2

**Author response: Done**

Line 325: Scenarios 84 and 90 are the top-ranked ones. How are they defined, and how do these two specific scenarios differ from the others? How does the numbering of scenarios work in Figure 2? In Figure 3A, which red box is scenario 84? And which is scenario 90?

**Author response:** We thank the reviewer for pointing out the need for clarification. The two best performing scenarios correspond to two top-performing combinations of soil water holding capacity parameters ( $AWC_{\text{mean}}$ ,  $AWC_{\text{min}}$ ) as shown in Figure 3A and 4A (highlighted with red boxes). The scenario numbering follows the internal index used in the parameter grid search (see Table S 2.1 in SM). We added these definitions and scenario labels directly to the figure captions 3 and 4 for clarity. We also clarified in both figure captions which box stands for which scenario.

Lines 324-328: Two significant digits are enough in percentage numbers.

Author response: Done, corrected across all manuscript.

Line 372: As in the previous comment, what are the features of the selected scenarios compared to the others? How does the numbering of scenarios work in Figure 2? In Figure 4A, which red box is scenario 76? And which is scenario 90?

**Author response:** See our response to the comment on 1. 325.

Lines 370-377: Two significant digits are enough in percentage numbers.

Author response: Done, , corrected across all manuscript.

Figure 3A and 4A: In both cases, scenario 90 is selected. However, figures 3A and 4A do not share the position of any red box. Figure 3A displays red boxes in row 1, columns 8 and 11; Figure 4A shows red boxes in row 1 column 9, and row 2 column 7. Check the red boxes in both figures.

**Author response:** Thanks to the reviewer's feedback we adjusted the wrong attribution of best selected scenarios to the corresponding box; we apologize for the oversight.

---

## Author Response (AR2)

**Revision of Manuscript egusphere-2025-1534**

In response to Reviewer 1 and the editorial request, we have revised the manuscript and Supplementary Material as follows:

1. *Additional falsification analysis and new figures in Supplementary Material*

    o Performed an additional falsification analysis in which AWC-scenario selection is based only on the 2000–2017 period.

    o Added the corresponding supplementary figures (for beech and spruce) that mirror the structure of Figs. 3 and 4 but restrict the comparison of observed vs. simulated mortality to 2000–2017.

    o Described these results in the revised Results section and expanded the discussion of scenario-selection uncertainty and period dependence.

2. *Clarified interpretation of pre-2018 mortality and model behavior*

    o Revised the Results and Discussion to clarify that pre-2018 mortality peaks are not purely stochastic but are linked to threshold-like model responses to moderate-to-severe drought conditions, while also highlighting cases where the model overestimates mortality.

    o Clarified the role of the drought index and medium drought conditions in triggering mortality under low AWC settings.

3. *Textual and structural clarifications in the main manuscript*

    o Corrected typographical and consistency issues pointed out by the reviewer 1

    o Regularized the Methods section numbering (including correcting the missing Section 2.2 and renaming Section 2.2.4 to "Simulation rationale and settings") and moved the statement about the model-intrinsic nature of the results to a more prominent position at the beginning of that section.

    o Added a clarification that the residuals were approximately Gaussian, justifying the Gaussian error assumption.

    o Implemented wording refinements suggested by the reviewer 1

4. *Clarification of PIC fractions and stochastic/background mortality*

    o Revised the caption and explanation of the relevant supplementary figure showing fractions of mortality attributed to PIC predisposing and inciting stress factors

5. *Revisions to the Supplementary Material and cross-references*

    o Renumbered all supplementary figures and tables consecutively (Fig. S1, S2, …; Table S1, S2, …), independently of section numbering.

o   Updated all references to supplementary figures and tables in the main text and Supplement to match the new numbering.

6. *Data and code availability*

o   Added and highlighted the DOI of the dataset (Marano and Bugmann, 2025) at its first mention and in the "Code and data availability" section.

A detailed, point-by-point response to all comments from Reviewer 1 is provided in the following section of this document.

**Author responses to Reviewer 1 of egusphere-2025-1534**

We thank the reviewer for the in-depth assessment of our work. We provide a point-by-point response below in red.

I highly appreciate the efforts the authors have taken in response to my first review. All in all, the additional analyses and textual refinements have improved the robustness of the manuscript which I deem almost ready for acceptance.

Yet, one relatively important aspect needs further consideration which refers to the selection of AWC-scenarios based on r² and MAE. In particular, the authors now provide more detailed insights into the behavior of r² and MAE in the period prior to 2018, indicating – as suspected – a lower fit between simulated and observed mortality rates. However, they explain this by stochasticity of mortality prior to 2018 which I do not see supported by some of the display items. That is, some of the mortality peaks prior to 2018 in fact seem to be caused by drought, wherefore a satisfying model fit in that period is desirable, too.

While this aspect does not question the overall mortality implementation at all (which I personally think is a strong point of the manuscript since not being based on calibration but model-internal mechanisms) it does have an impact on the presented results. To make this clear early on: I am not asking for major revisions of the results section. But I would like to see a few more aspects of falsification and visualization of uncertainty in the supplementary alongside a brief discussion/evaluation of these items in the results and discussion. This basically includes two figures comparable to Figs. 3 and 4 but only for the calibration period 2000-2017. This would clearly visualize, how a different period used for the identification of the best AWC-scenarios would affect the selection and consequently the simulated mortality rates. The authors can then briefly mention these effects in the corresponding results sections (lines 364-370 for beech and 438-444 for spruce) and elaborate the discussion in lines 479-486 and/or 572-576. Doing so, allows readers to fully capture the effects of period selection on model selection and the related results.

To make clear, why I think this point is so important I would like to bring up a hypothetical scenario: Imagine, the study were done in 2017 and you would only have the observational data available until that year. Back then, the selected AWC-scenarios would be different (as indicated by Fig. S 3.3.1). It would be interesting to see how these other models predict the mortality after 2017, i.e. outside the period that was used as a baseline for selecting these models as best fit. This hypothetical scenario stands example for the current situation, where we do not know what

will happen in the future. So examining this hypothetical example would help to assess the uncertainty related to the AWC-scenario selection approach and related interpretations.

We thank the reviewer for these insightful and constructive reflections and comments. We agree that this evaluation strengthens the transparency and credibility of the scenario selection process, particularly under the hypothetical condition where the future (post-2017) is unknown. Therefore, we have conducted a falsification analysis as requested, using only the 2000–2017 period for model evaluation and scenario selection. The results of this analysis are now included in the Supplementary Materials (Figures S10 and S3). These figures mirror the format of Figures 3 and 4 but limit the observed vs. simulated mortality comparison to the pre-2018 period. This approach, together with Figures S 3.2.1.1 and S 3.2.2.2 respectively, allows for a clearer visualization of how the AWC scenario selection would differ if only earlier data had been available.

For beech (Fig. S10), this led to the selection of scenarios 1 and 4 (Fig. 10, D-E), both representing very low AWC conditions (AWCmean = 7.5 cm; AWCmin = 7.5 and 15 cm, respectively; see red boxes in Figure S10, A). This is consistent with the model's stronger drought sensitivity under low AWC (Figure S8), with mortality peaks occurring in known drought years (e.g., 2003–2004 and 2012), as shown in Figure S9, A. We acknowledge the reviewer's point that these peaks reflect a response to drought rather than stochasticity, and we have revised the manuscript (lines 371-376) accordingly to clarify that the observed mortality prior to 2018 is not random but linked to threshold-like model behavior in response to moderate-to-severe drought conditions. However, there are also years such as 2015–2016 (scenario 4) where the model simulates mortality spikes not supported by observations (Figure 1, main text). These may result from an overreaction to the simulated drought intensity and combined with background mortality events, in the absence of observed mortality peaks (lines 376-378).

For spruce (Fig. S13), the scenarios selected under the 2000–2017 window were scenarios 29 (AWCmean = 15 cm, AWCmin = 12.5 cm) and 41 (AWCmean = 17.5 cm, AWCmin = 15 cm) (see red boxes in Figure S13, A and panels D-E). These closely align with the full-period selection (where scenario 42 was preferred over 41), suggesting a more stable scenario ranking for spruce irrespective of the calibration period. The performance metrics for these scenarios (Adj $R^2$ = 0.258 and 0.212; MAE = 0.733% and 0.631%, respectively) indicate reasonable consistency despite slightly lower observed mortality and weaker drought signals in the earlier years. Similar to beech, stronger drought years tend to trigger mortality more readily under low AWC, although in some cases—such as 2003–2004, and especially 2010–2012 and 2015–2016—the model overshoots, simulating mortality peaks not evident in the observations. This behavior reflects persistent medium drought index values (DI > 0.1–0.15) during those years (Figure S12, A), which drive the model response, particularly under low AWC settings.

Overall, this additional analysis confirms the qualitative consistency of model behavior across periods: low AWC scenarios lead to early and often elevated drought-induced mortality for both species. However, it also highlights the sensitivity of scenario selection to the calibration window, particularly for beech. We now explicitly discuss this in the revised Results section (lines 371-378 for beech; lines 451–455 for spruce) and in the Discussion section (lines 493–497 and 592–595). Notably, spruce scenario rankings were more temporally stable than those

of beech, pointing to species-specific differences in how robustly model selection captures mortality dynamics across time.

We agree with the reviewer that model evaluation under both moderate and extreme drought conditions is essential, especially if the model is to be used for future projections. Our additional analysis therefore contributes to a more balanced assessment of scenario uncertainty, which does not only focus on the post-2018 period.

Apart from this, I only have a few minor points as outlined below.

Line 16: 'four' hypotheses is probably a legacy from the initial submission. I guess it should be three

We have corrected this at line 16.

Line 37: it seems the word 'of' is missing in this sentence: understanding, forecasting and managing 'of' forest resistance

We have corrected this at line 37.

Lines 111 and 122: it is justified to only select the sites from Knapp et al. (2024) but I nevertheless suggest to reflect in the text why northeastern Germany is basically empty, since readers may wonder (as I did) why that is the case.

We have added a consideration about the lack of sites in N-E Germany at lines 113-116.

It seems that section 2.2 is missing, please adjust (I guess section 2.3 should be 2.2)

We apologize for this oversight. We have corrected all numbering accordingly.

Lines 232 and 234: BGB has not been introduced before. Or do the authors refer to BGR which was mentioned in line 225? Please clarify.

We apologize for this glitch; we referred to the BGR. We have corrected this at lines 235 and 237.

Lines 256-260: I wonder whether the authors want to find a more prominent spot for mentioning this important point, e.g. at the beginning of section 2.3 (which should be 2.2. I guess) to stress that the results represent model-intrinsics and not empirical fits which I personally evaluate as a major strength of the presented approach.

We have moved this consideration to the beginning of section 2.2.4, which we renamed to "Simulation rationale and settings" (lines 214-219).

Line 282: I understand from your reply that your data were quasi-normal distributed wherefore you assumed a Gaussian fit. Just for the sake of clarity, I recommend to add this information here since readers may have similar doubts as I did.

We thank the reviewer for this remark and have decided to add a clarifying statement at lines 285-286.

Lines 364-370: I highly appreciate that the authors have done this additional analysis to reflect the model-performance skills based on a subset of the data. Yet, I do not agree with the conclusion that most of the observed variance before 2018 originates from stochasticity which also contrasts some of the statements in the results section (e.g. lines 350 and 373). Inspection of Fig. S 3.2.1.2 clearly shows, that the years 2003, 2012, and 2015 were characterized by a higher drought index which is mirrored in increased mortality rates in both species. Thus, the peaks of mortality rates prior to 2018 do not originate from stochasticity but an increasing DI which also becomes evident if inspecting Fig. S 3.3.2. As I see it, this mirrors the threshold-like behavior of the mortality model which – as the authors conclude themselves – results in an overestimation of mortality for beech in general and for spruce in the years prior to 2014. Moreover, I would not call the increase of $r^2$ an artifact but rather a mathematical feature of the calculation of correlations. The mortality rates following 2017 increase the error squares by an order of magnitude and if this happens in both simulated and observed mortality, $r^2$ (and MAE) will increase substantially. Since your best AWC-scenario selection Is based on $r^2$ (and MAE), this is a crucial aspect to consider. Based on Fig. S 3.3.1, it seems that if the scenario selection had been based on the years prior to 2018, different scenarios were selected. The question then arises, how these scenarios performed in the years after 2017. I am not asking the authors to change the scenario selection for the main text at this point, but I strongly recommend to reflect this behavior more clearly in the results and discussion (see also my main comment above). Ideally, the authors would show two supplementary display items, resembling Figs. 3 and 4 but for the shorter calibration period 2000-2017. My concern is that if at any point the aim is to use such kind of mortality models for projections into the future, I would rather rely on AWC-scenarios whose $r^2$ also performs acceptable under less extreme conditions than 2018. Otherwise, the selected AWC-scenarios might result in too extreme mortality scenarios. Since a model validation based on a subset of data is a common standard, I see the reflection of uncertainty of model performance metrics ($r^2$ and MAE) as a mandatory point to reflect in terms of clarity.

We thank the reviewer for his comment. As motivated in the answer above, we modified our statement in both the Results (lines 371-378 for beech; lines 451–455 for spruce) and in the Discussion (lines 493–497 and 592–595) sections.

Line 378: to ease the reading I suggest to add 'combined' before 'long- and short' or revise into: the interplay of effects from drought-effects acting on longer and shorter time-scales.

We have accepted the suggestion and modified the text accordingly at line 382.

Lines 438-444: see my comment on lines 364-370. For spruce, it is also interesting to note, that the increasing observed mortality after 2012 is not captured by the model. Based on the higher complexity of the AWC-scenario performance of Norway spruce I am even more curious to know what would happen if selecting the models based on performance metrics derived from the period 2000-2017. Again, I am not asking the authors to change the key display items of the results section but I would highly appreciate a more comprehensive model falsification (see my suggestions above).

We provide a detailed answer on spruce behavior for the falsification experiment in the detailed answer above.

Lines 450-458: I highly appreciate the inclusion of this sensitivity analysis. The new results totally make sense wherefore they strengthen your model implementation.

We thank the reviewer for this feedback.

Line 468: While I generally agree with this statement, I want to highlight that roughly one third of the mortality observed in 2019 stemming from PIC arises from inciting factors. Also, I am puzzled about the fractions presented in Fig. S 3.3.2. I assume, that total mortality fraction adds up to 100% in each year. So, is the remaining part (which is not shown but adds up to 100%) stochastic mortality? At least I don't fully get how this SI-figure goes together with the main display items which show different temporal dynamics of the mortality rates. Could the authors please elaborate on this? Maybe this also relates to your interpretation of stochasticity dominating the mortality before 2018 so getting this right is quite essential.

We thank the reviewer for pointing this out. We clarify that Figure S 3.3.2 does not represent the full annual mortality ($T_{mortality}$) summing to 100%. Instead, it shows the fraction $\bar{T}_{PIC}$(% of dead trees per year) with DBH $\geq 40$ cm, flagged for predisposing and inciting stress factors in the PIC framework. Hence, it only includes the subset of dead trees for which a mechanistic cause could be assigned (slow growth, drought memory, or inciting drought stress).

The remaining proportion of dead trees not accounted for in this figure stems from (i) other size classes, and (ii) trees that died without meeting PIC thresholds, i.e., considered as background/stochastic mortality. This explains why the total bars do not sum to 100%.

We have now clarified this in the figure caption (lines 112-114) to avoid misinterpretation. This distinction also aligns with our interpretation that before 2018, much of the mortality arose from stochastic processes, while the post-2018 mortality increasingly involved trees meeting specific drought-related stress thresholds.

Lines 479-486: I appreciate this critical reflection of the newly added subsample-analysis. Yet, I again want to stress that the patterns before 2018 do not only arise from stochasticity but appear to be triggered by drought events (e.g. 2012, 2015). If implementing my suggestion of additional supplementary display items resembling Figs. 3 and 4 but for the period 2000-2017, the authors could elaborate the discussion in this section to reflect how the period selection affects the AWC-scenario selection. This would also guide future research that mimics the presented approach and address aspects of uncertainty related to a mortality projections based on the AWC-scenarios. I understand that this is not the main point of the manuscript, but it will help to correctly guide related future work.

We appreciate once again the reviewer's comment. We have integrated these considerations in the Discussion section, lines 493–497 and 592–595, respectively.

Lines 572-576: I highly appreciate this paragraph to foster cautious reproduction of the AWC-scenario approach. Maybe the points I suggested for lines 479-486 could instead be added here.

We have added these considerations in the Discussion section lines 493–497 and 592–595, respectively.

Line 586: please check this sentence: 'but it does induce longer die-off once water availability improves'. This does not make sense in combination with the following sentence. Or is it simply the word 'die-off' which is ambiguous in context of mortality discussions? That is, the stress dies off and not the trees? Please clarify and revise.

We thank the reviewer for pointing out the ambiguity in this sentence. Our intention was to describe the persistence of physiological stress symptoms even after drought conditions subside, not a continued increase in tree mortality. We have revised the sentence accordingly to avoid confusion with the term "die-off" at lines 590-591.

---

## Author Response (AR4)

**Author's response to Reviewer 1**

We thank Reviewer 1 for their careful re-evaluation of the revised manuscript and for the positive assessment of the changes made so far.

The reviewer noted that in Supplementary Figures S10 and S13 the relationship between observed and simulated mortality had become negative, and correctly suggested that this could arise because the selection of best-performing AWC scenarios was based on goodness-of-fit metrics that do not explicitly constrain the sign of the relationship. We agree with this assessment.

In response, we revised the AWC scenario-selection procedure by introducing an additional constraint such that only scenarios exhibiting a positive observed–simulated mortality relationship are eligible for selection. Within this constrained subset, scenarios are then ranked using the same performance criteria as before (adjusted $R^2$ and MAE). This prevents the selection of scenarios with inverted relationships, even if their goodness-of-fit metrics are comparatively high.

Supplementary Figures S10 and S13 have been updated accordingly. All selected scenarios shown in these figures now exhibit positive observed–simulated mortality relationships, ensuring a meaningful and interpretable comparison between observations and simulations. The corresponding interpretation in the main text was updated to reflect the revised selection outcome (lines 371-376).

**Editorial and supplementary material corrections**

In addition, we have addressed the editorial office's comments regarding the supplementary material. Supplementary figures and tables are now numbered consecutively (Figure S1, Figure S2, Table S1, etc.), independent of section numbering.